# Live imaging of excitable axonal microdomains in ankyrin-G-GFP mice

Christian Thome[1,2,3], Jan Maximilian Janssen[1,2,4], Seda Karabulut[4], Claudio Acuna[5], Elisa D'Este[6], Stella J Soyka[7], Konrad Baum[1,2], Michael Bock[4], Nadja Lehmann[4], Johannes Roos[1,2,4], Nikolas A Stevens[3], Masashi Hasegawa[8], Dan A Ganea[9], Chloé M Benoit[8,9], Jan Gründemann[8,9], Lia Y Min[10], Kalynn M Bird[10], Christian Schultz[4], Vann Bennett[11], Paul M Jenkins[10,12]*, Maren Engelhardt[1,2,4]*

[1]Institute of Anatomy and Cell Biology, Johannes Kepler University, Linz, Austria; [2]Clinical Research Institute for Neurosciences, Johannes Kepler University, Linz, Austria; [3]Institute of Physiology and Pathophysiology, Heidelberg University, Heidelberg, Germany; [4]Institute of Neuroanatomy, Mannheim Center for Translational Neuroscience (MCTN), Medical Faculty Mannheim, Heidelberg University, Mannheim, Germany; [5]Chica and Heinz Schaller Research Group, Institute of Anatomy and Cell Biology, Heidelberg University, Heidelberg, Germany; [6]Optical Microscopy Facility, Max Planck Institute for Medical Research, Heidelberg, Germany; [7]Institute of Anatomy and Cell Biology, Dept. of Functional Neuroanatomy, Heidelberg University, Heidelberg, Germany; [8]German Center for Neurodegenerative Disease (DZNE), Neural Circuit Computations, Bonn, Germany; [9]Department of Biomedicine, University of Basel, Basel, Switzerland; [10]Department of Pharmacology, University of Michigan Medical School, Ann Arbor, United States; [11]Department of Biochemistry, Duke University Medical Center, Durham, United States; [12]Department of Psychiatry, University of Michigan Medical School, Ann Arbor, United States

*For correspondence:
pjenkins@umich.edu (PMJ);
maren.engelhardt@jku.at (ME)

**Competing interest:** The authors declare that no competing interests exist.

## eLife Assessment

In this **valuable** paper, the authors created a reporter mouse line in which the Axon Initial Segment (AIS) is intrinsically labeled by an ankyrin-G-GFP fusion protein activated by Cre recombinase, tagging the native *Ank3* gene. Using confocal, superresolution, and two-photon microscopy as well as whole-cell patch-clamp recordings in vitro, ex vivo, and in vivo, the authors **convincingly** document that the subcellular scaffold of the AIS and electrophysiological parameters of labeled cells remain unchanged. They further uncover rapid AIS remodeling following increased network activity in this model system, as well as highly reproducible in vivo labeling of AIS over weeks.

**Abstract** The axon initial segment (AIS) constitutes not only the site of action potential initiation, but also a hub for activity-dependent modulation of output generation. Recent studies shedding light on AIS function used predominantly post-hoc approaches since no robust murine in vivo live reporters exist. Here, we introduce a reporter line in which the AIS is intrinsically labeled by an ankyrin-G-GFP fusion protein activated by Cre recombinase, tagging the native *Ank3* gene. Using confocal, superresolution, and two-photon microscopy as well as whole-cell patch-clamp recordings in vitro, ex vivo, and in vivo, we confirm that the subcellular scaffold of the AIS and electrophysiological parameters of labeled cells remain unchanged. We further uncover rapid AIS remodeling following increased network activity in this model system, as well as highly reproducible in vivo labeling of AIS over weeks. This novel reporter line allows longitudinal studies of AIS modulation

and plasticity in vivo in real-time and thus provides a unique approach to study subcellular plasticity in a broad range of applications.

## Introduction

The AIS is the cellular compartment where most neurons integrate synaptic input and generate their primary output signal, the action potential (AP); (*Bender and Trussell, 2012*; *Kole and Stuart, 2012*). Recently, technological advances have shed new light on this critical axonal microdomain, in particular its structure, molecular composition, and regulation.

The AIS constitutes a central site for the regulation of neuronal output. It is critical for the initiation of APs and receives input from specialized inhibitory neurons which enables synchronization of neuronal firing across nearby cells (*Wefelmeyer et al., 2016*). Much like synapses, the AIS responds to changes in neuronal activity with different types of homeostatic plasticity, defined by structural changes in AIS length, position, and channel architecture (*Jamann et al., 2021*; *Grubb and Burrone, 2010*; *Kuba et al., 2010*; *Zbili et al., 2021*). Most recent studies investigating AIS function and plasticity relied on ex vivo immunofluorescent staining in fixed neuronal tissue and cultures using antibodies against elements of the AIS cytoskeleton and post-hoc analysis. These targets make sense, since the scaffolding proteins ankyrin-G and βIV-spectrin cluster voltage-gated ion channels at the AIS and provide a cellular anchor between the axonal membrane and cytoskeleton (*Jenkins et al., 2015*; *Leterrier et al., 2015*; *Leterrier et al., 2017*). Recent approaches allow for live labeling of the AIS using antibodies against the extracellular domain protein neurofascin-186 (*Dumitrescu et al., 2016*) or by transfecting cultured primary neurons with ankyrin-G-GFP constructs (*Komada and Soriano, 2002*). However, these techniques are limited in their application due to challenges imposed by the AIS itself. For example, ankyrin-G and βIV-spectrin are large proteins that are, therefore, difficult to express (*Jenkins et al., 2015*; *Komada and Soriano, 2002*; *Berghs et al., 2000*) and their overexpression, when successful, may conceivably have an altering impact on AIS architecture and/or function. Also, all currently available live reporters are solely established for in vitro applications. Other studies uncovering the role of the AIS in the modulation of neuronal excitability combined electrophysiological recordings with post-hoc immunostainings (*Grubb and Burrone, 2010*; *Kuba et al., 2010*; *Wefelmeyer et al., 2015*; *Kim et al., 2019*), preventing longitudinal tracking of AIS plasticity in the same cells over time.

To overcome these limitations, we have generated a mouse line for Cre-dependent expression of an ankyrin-G-GFP fusion protein using the native *Ank3* locus. This novel mouse line allows the monitoring of AIS parameters in vitro (isolated cells, organotypic cultures, OTC), ex vivo (acute brain slices), and in vivo (living mice). Using this model, we demonstrate efficient labeling of AIS and nodes of Ranvier (noR) in vitro, ex vivo, and in vivo by expression of Cre recombinase. The Cre-dependence of this model allows labeling of the AIS and noR in genetically defined neuronal populations and with temporal precision in the living animal or acute slice, depending on the selected Cre-line or Cre-supplementing virus. We show that ankyrin-G-GFP mice have normal AIS properties, including clustering of binding partners, inhibitory synaptic clustering, and nanostructure. In addition, we demonstrate that the electrophysiological properties of ankyrin-G-GFP neurons are indistinguishable from those found in wild-type littermate controls, including AP properties and excitatory synaptic function. In a proof-of-principle approach, we further demonstrate AIS plasticity in OTC from ankyrin-G-GFP mice. Finally, we show that AIS can be imaged in vivo with two-photon microscopy in both superficial and deep brain regions and that the ankyrin-G-GFP-labeled AIS can be imaged over days and weeks. Thus, our model allows the monitoring of ankyrin-G-dependent structures, like the AIS and noR, in neuronal cultures, acute slices, and intact animals, providing a much-needed tool for the visualization and analysis of axonal plasticity in vivo.

## Results

### Intrinsic AIS labeling in various experimental in vitro, ex vivo, and in vivo systems

Ankyrin-G is an intracellular scaffolding protein critical for the formation and maintenance of the AIS, where it recruits multiple cytoskeletal elements, ion channels, transporters, and cell adhesion molecules that are necessary for normal AIS function (*Nelson and Jenkins, 2017*; *Rasband, 2010*; *Leterrier, 2018*). We, therefore, decided to use ankyrin-G as a backbone to design a conditional mouse model to intrinsically label the AIS (see Methods). Ankyrin-G tolerates the addition of a GFP tag, since ankyrin-G-GFP can rescue multiple functions of ankyrin-G in *Ank3* null neurons, including scaffolding of voltage-gated sodium and potassium channels, clustering of the spectrin cytoskeleton, and recruitment of inhibitory synapses (*Jenkins et al., 2015*; *Fréal et al., 2016*; *Tseng et al.,*

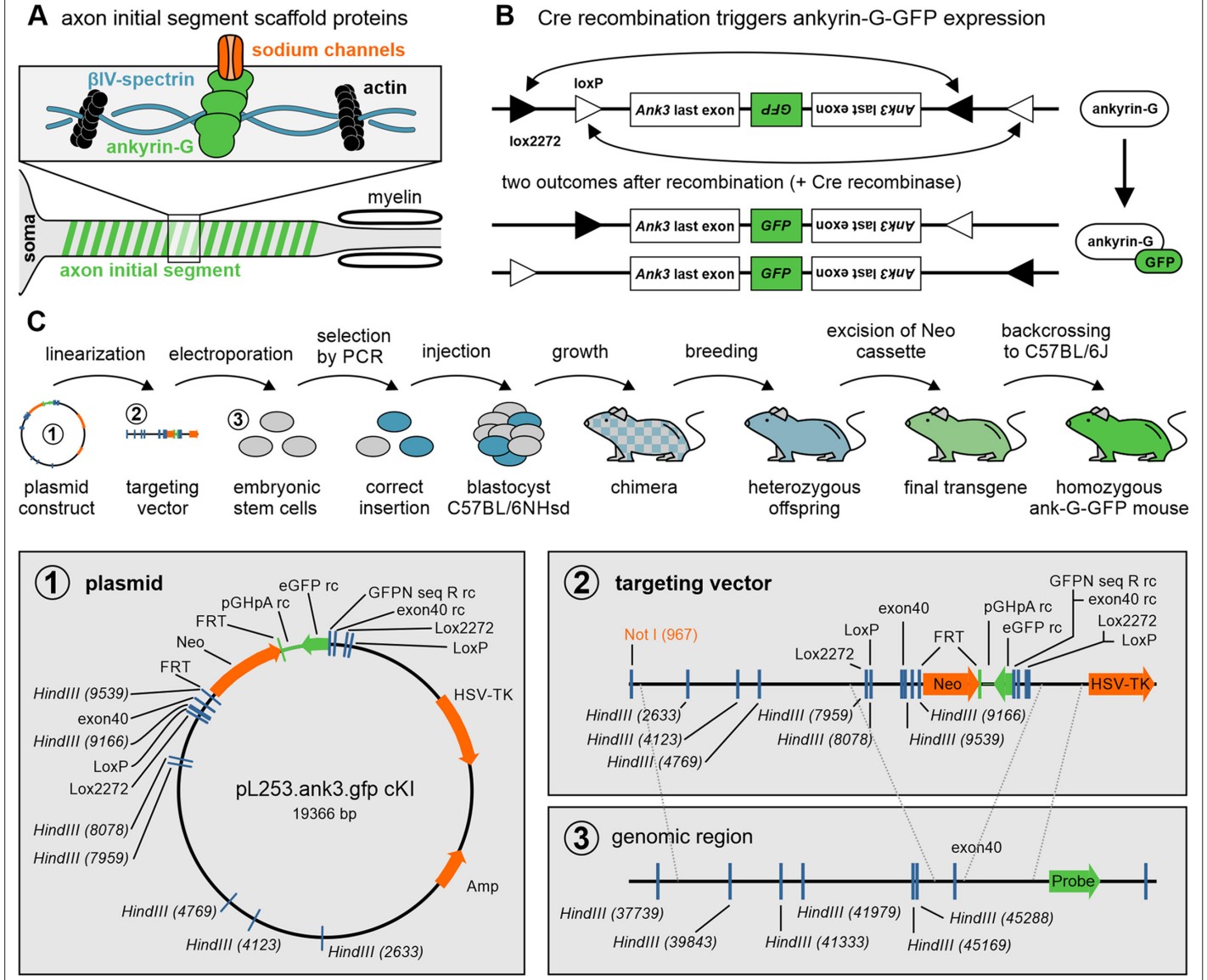

**Figure 1.** Genetic strategy and creation of the ank-G-GFP line. (**A**) Sketch of the intracellular scaffold underlying the axon initial segment (AIS). Ankyrin-G anchors ion channels to the axonal membrane. Ankyrin-G itself has spectrin binding domains, thus interacting with βIV-spectrin rings, which in turn bind to the axonal actin cytoskeleton. (**B**) Outline of transgenic modification before and after Cre-dependent recombination of ankyrin-G. The arrangement of the loxP and lox2272 sites ensures that recombination occurs only once. Both possible outcomes produce a functional Ank3 last exon with a coupled GFP sequence. (**C**) Sketch of the sequence of operations to create the ankyrin-G-GFP line. Detailed steps are outlined in the method section. The plasmid, targeting vector, and genomic target region are enlarged in the lower panel.

*2015*). To intrinsically label ankyrin-G with GFP, we utilized a FLEx cassette system to flip out the last exon of *Ank3* and replace it with the last exon of *Ank3* fused with the coding sequence for GFP (*Figure 1*). Given the heterogeneous expression of ankyrin-G in various cell compartments across multiple tissues, constitutive tagging of the endogenous *Ank3* allele would make it difficult to visualize ankyrin-G-dependent structures. By contrast, our system allows us to only GFP-label ankyrin-G in cells expressing Cre recombinase, allowing the visualization of ankyrin-G in genetically and temporally defined cell populations.

When generating GFP fusion proteins, it is critical to confirm that GFP does not interfere with normal protein function. To this end, we utilized the linker-GFP sequence from the rescue plasmids used in multiple studies from our group and others to examine the function of the three main classes of brain ankyrin-G isoforms, 190, 270, and 480 kDa, respectively. Importantly, these GFP fusions have been shown to be sufficient to rescue *Ank3* knockout or silencing in the formation of the AIS, noR, GABAergic synapses, and dendritic spines (*Jenkins et al., 2015*; *Fréal et al., 2016*; *Tseng et al., 2015*; *Smith et al., 2014*).

To characterize the ankyrin-G-GFP (ank-G-GFP) mouse, we utilized several of the most common recombination systems and experimental paradigms used in cellular neuroscience and confirmed intrinsic AIS features in different neuronal cell types in vitro, ex vivo, and in vivo across multiple brain regions (*Figure 2A*). Ank-G-GFP mice were first used to generate hippocampal OTC, which were then exposed to AAV expressing Cre-recombinase under the control of the synapsin promotor (AAV5-pmSyn1-EBFP-Cre). After 48 hr, neurons exposed to the virus showed GFP expression along their AIS (*Figure 2B*, left panel). We confirmed the GFP signal as originating from AIS by staining against βIV-spectrin (*Figure 2B*, middle panel). Not all AIS in OTC neurons were GFP$^+$, likely due to the transfection efficiency with AAV-Cre. By directly injecting viruses, and by crossbreeding ank-G-GFP mice with different Cre driver lines, we overcame this limitation and used acute slices or cryosections to verify efficient labeling of the AIS. The injection of ank-G-GFP animals with a synapsin-Cre-tdTomato AAV resulted in ank-G-GFP$^+$-AIS (*Figure 2C*, left panel), which again were βIV-spectrin$^+$ as well (*Figure 2C*, middle panel). Of note, the somatic envelope and proximal dendrites were also visible via the ank-G-GFP signal (*Figure 2C*, asterisks in left panel), highlighting ankyrin-G expression in the somatodendritic compartment of mature neurons (*Tseng et al., 2015*; *Nelson et al., 2020*). Likewise, using a combination injection of AAV1-hDlx-Flex-dTomato-Fishell_7 and AAV1-hSyn.Cre.WPRE.hGH, allowed visualization of ank-G-GFP$^+$-AIS in hippocampal interneurons via tdTomato expression (*Figure 2D*).

Mating of ank-G-GFP mice with a line expressing Cre recombinase under the control of the Calcium/calmodulin-dependent protein kinase type II alpha chain (CaMKIIα) promoter resulted in reproducible reporter detection in CaMKII-expressing neurons in layer II neocortical pyramidal neurons (*Figure 2E*). Neurons not expressing CaMKII were negative for ank-G-GFP, but as expected, additional staining against βIV-spectrin (*Figure 2E*, middle panel) resulted in labeling of all AIS in the section, allowing for clear distinction between the different populations (ank-G-GFP$^+$-AIS vs ank-G-GFP$^-$-AIS).

Ank-G-GFP expression is not limited to typical cortical neurons (pyramidal cells, interneurons). We found reliable ank-G-GFP expression in retinal whole-mount preparations, both in peripheral and central retinal ganglion cells (RGC; *Figure 2F*; ank-G-GFP x CaMKII-Cre animals). Colabeling of ank-G-GFP$^+$ AIS with antibodies directed against native ankyrin-G in RGC indicated that the vast majority of RGC express CaMKII and consequently, their AIS are positive for both GFP and the intrinsic AIS marker ankyrin-G (*Figure 2F*).

Interneuron populations showed similar selective labeling when using appropriate Cre driver lines. Mating of ank-G-GFP animals with parvalbumin (PV)-Cre mice highlighted exclusive GFP signal in AIS that belonged to PV$^+$ interneurons as shown in ex vivo acute slices of the cerebellum (*Figure 2G*, left panel). Again, double labeling against βIV-spectrin indicated distinct GFP expression only in those neurons that expressed PV (*Figure 2G*, middle panel). In all experiments outlined above, double labeling against the characteristic AIS markers βIV-spectrin and ankyrin-G, respectively, served as control of AIS identity of labeled structures and we found that ank-G-GFP reporter expression always colocalized with these markers.

## Ank-G-GFP localizes to nodes of Ranvier

The 480 kDa isoform of ankyrin-G is also localized to nodes of Ranvier (noR) in myelinated axons where it serves as a scaffold for voltage-gated ion channels similar to those found at the AIS (*Jenkins*

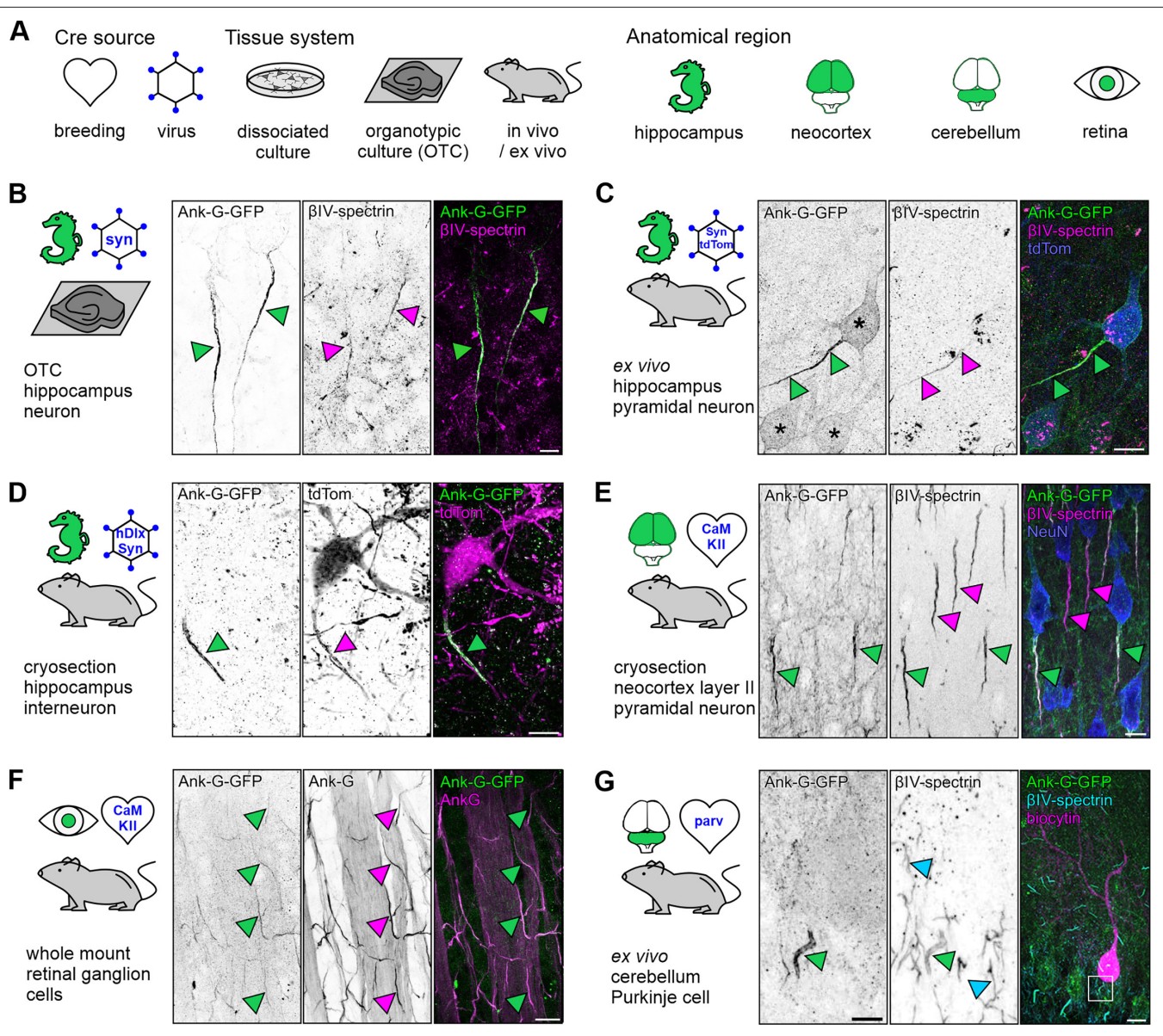

**Figure 2.** Ank-G-GFP activation and expression in different experimental models intrinsically highlights the AIS in distinct neuron populations. (**A**) Experimental design. Ank-G-GFP expression was triggered by (**i**) exposure of organotypic slice cultures (OTC) to an AAV-Cre, (ii) injection of AAV-Cre into ank-G-GFP mice, or (iii) breeding of ank-G-GFP mice to different Cre-lines. Anatomical regions are highlighted in green. Cartoons are used throughout the manuscript to indicate the brain regions investigated, the procedure of Cre-activation (heart = breeding with Cre-line; hexagon = AAV infection in vitro or injection in vivo), and tissue types (petri dish = dissociated cells; brain slice = OTC; mouse = cryosection, ex vivo acute slice, and in vivo). (**B**) Hippocampal OTC at days in vitro (DIV) 14 from ank-G-GFP mice exposed to AAV expressing Cre-recombinase under the synapsin promotor (AAV5-pmSyn1-EBFP-Cre). Ank-G-GFP+-AIS (left, green arrowheads) colabeled with βIV-spectrin (middle, magenta arrowheads) in pyramidal neurons; merged channels (right). Scale bar = 5 μm. (**C**) Cryosection prepared from an ank-G-GFP animal injected with a synapsin-Cre-tdTomato AAV 3 wk after injection. The ank-G-GFP+AIS (left, green arrowheads) is clearly discernible. Immunostaining against βIV-spectrin (middle, magenta arrowheads) indicates the same axon initial segment (AIS) (merged, right). Note the ank-G-GFP+ somatic envelop of hippocampal excitatory neurons (left, asterisks). Scale bar = 10 μm. (**D**) An ex vivo acute slice prepared from the hippocampus of an ank-G-GFP mouse injected with a combination of AAV1-hDlx-Flex-dTomato-Fishell_7 and AAV1-hSyn.Cre.WPRE.hGH. The tdTomato-positive interneuron (middle and right, magenta) is endowed with an ank-G-GFP+-AIS (left, green arrowhead). Scale bar = 10 μm. (**E**) Cryosection of neocortex from an ank-G-GFP × CaMKIIa-Cre mouse, highlighting ank-G-GFP+-AIS (left and middle, green arrowheads) and ank-G-GFP-, but βIV-spectrin+AIS (middle, magenta arrowheads) in layer II pyramidal neurons; merged channels (right). Scale bar = 10 μm. (**F**) Retinal whole mount preparation from an ank-G-GFP × CaMKIIa-Cre mouse. The image shows a peripheral aspect of the retina. The overlap of ank-G-GFP + AIS with the colabeling of ank-G is evident (green and magenta arrowheads in all panels), indicating that likely all RCG express CaMKII and consequently, all AIS are positive for both GFP and the intrinsic AIS marker ankyrin-G. Scale bar = 20 μm. (**G**) A cerebellar ex vivo acute slice prepared from ank-G-GFP × PV-Cre mice. This Purkinje cell was filled with biocytin via a patch pipette and stained with

*Figure 2 continued on next page*

*Figure 2 continued*

Streptavidin (right, magenta). The ank-G-GFP + AIS (left, green arrowhead, magnified from boxed region in right panel) is clearly discernible from surrounding ank-G-GFP-, βIV-spectrin + AIS (middle, cyan arrowheads). Scale bar (left and middle) = 10 μm, right = 20 μm.

*et al., 2015*; *Jenkins and Bennett, 2002*; *Dzhashiashvili et al., 2007*). Consequently, the ank-G-GFP construct should also be discernible in this axonal microdomain. In mice expressing ank-G-GFP in CaMKIIa-positive excitatory neurons, noR in the alveus of hippocampal CA3 were visible as distinct puncta that co-localized with other nodal markers such as ankyrin-G and Na$_v$1.6 (*Figure 3A*). Of note, only CaMKII-expressing neurons showed ank-G-GFP signals in noR (*Figure 3A*, green circles). Ank-G-GFP⁻ noR were still characterized by the colocalization of ankyrin-G and Na$_v$1.6 (*Figure 3A*, blue circles). To confirm that the expression of ank-G-GFP does not disrupt the nodal structure, we examined the localization of the paranodal Contactin-associated protein (Caspr) (*Rios et al., 2000*). Immunostaining against GFP, ankyrin-G, and Caspr revealed normal localization of ank-G-GFP and Caspr to the node and paranode, respectively, in myelinated axons of CaMKII-positive neurons (*Figure 3B*). The quantification of node shape and fluorescence signals of Na$_v$1.6 and ankyrin-G showed no differences between GFP⁺ and GFP⁻ nodes (*Figure 3C*, *Figure 3—figure supplement 1A and B*). We note that these groups belong to genetically distinct subgroups of cells, but consistent node lengths and Na$_v$1.6 levels indicate that the GFP label does not grossly disrupt node structure. Furthermore, GFP label intensities within GFP⁺ nodes did not interfere with Na$_v$1.6 levels (*Figure 3—figure supplement 1C*, top panel). Instead, we found a strong correlation between GFP and ankyrin-G in GFP⁺ nodes, demonstrating the specificity of our model (*Figure 3—figure supplement 1C*, bottom panel).

## Ank-G-GFP expression preserves axonal characteristics and does not alter the molecular composition of the AIS

The AIS and its niche are characterized by a number of intra- and extracellular components contributing to its unique structure and function (*Leterrier, 2018*). To investigate whether the fusion of GFP to endogenous ankyrin-G has any impact on core AIS features, we selected additional targets for immunolabeling (*Figures 4 and 5*) and compared whether they are affected by the GFP modification. We found that the length and distance of AIS (measured by βIV-spectrin signal) did not differ between neighboring GFP⁺ and GFP⁻ CA1 pyramidal cells (*Figure 4—figure supplement 1A*). Length measurements using the live ankyrin-GFP label in combination with post-fixed ankyrin-G or βIV-spectrin of the same AIS showed a strong correlation with one another (*Figure 4—figure supplement 1B–C*).

A major feature of the AIS is its endowment with voltage-gated ion channels, the anatomical prerequisite for proper AP generation and propagation (*Kole et al., 2008*; *Kole et al., 2007*; *Bender and Trussell, 2009*). Colabeling of the sodium channel isoform Na$_v$1.6 with βIV-spectrin in ank-G-GFP⁺ AIS in cortical layer V pyramidal neurons showed homogeneous signals of Na$_v$1.6 across the length of neighboring GFP⁺ and GFP⁻ AIS. Note that previously published data found a preference of Na$_v$1.6 for distal expression at the AIS in the layer V prefrontal cortex (instead of layer II somatosensory cortex; *Van Wart et al., 2007*; *Van Wart et al., 2007 Figure 4A*, *Figure 4—figure supplement 1D*). Likewise, a similar expression of K$_v$2.1 was observed in GFP⁺ and GFP⁻ AIS (*Figure 4B*, *Figure 4—figure supplement 1E*).

The AIS is also characterized by the expression of several cell adhesion molecules, chief among them neurofascin-186 (NF-186), which is important for the recruitment of ion channels to the AIS during development (*Hedstrom et al., 2007*; *Alpizar et al., 2019*). Potential disruption of normal AIS assembly in our reporter model may have an impact on the assembly of the extracellular niche at the AIS, however, immunostaining against NF-186 in ank-G-GFP⁺ AIS produced the expected co-staining pattern (*Figure 4C*). The AIS-related proteins TRIM46, organizing the fasciculation of microtubules in the AIS (*Fréal et al., 2019*; *Harterink et al., 2019*) and FGF14, regulating voltage-gated ion channels at the AIS (*Pablo and Pitt, 2017*), were also detected in ank-G-GFP⁺ neurons throughout the neocortex of ank-G-GFP x CaMKII-Cre mice (*Figure 4D and E*).

Next, we investigated whether the cisternal organelle (CO), an AIS-specific intra-axonal Ca²⁺-store, remained unperturbed by ank-G-GFP expression. The actin-binding protein synaptopodin (synpo) has been shown to be a reliable and reproducible marker for the CO in the AIS (*Hanemaaijer, 2020*; *Schlüter et al., 2017*; *Sánchez-Ponce et al., 2012*). Immunostaining with antibodies against GFP, synpo, and βIV-spectrin in layer V pyramidal neurons of primary motor cortex of ank-G-GFP x

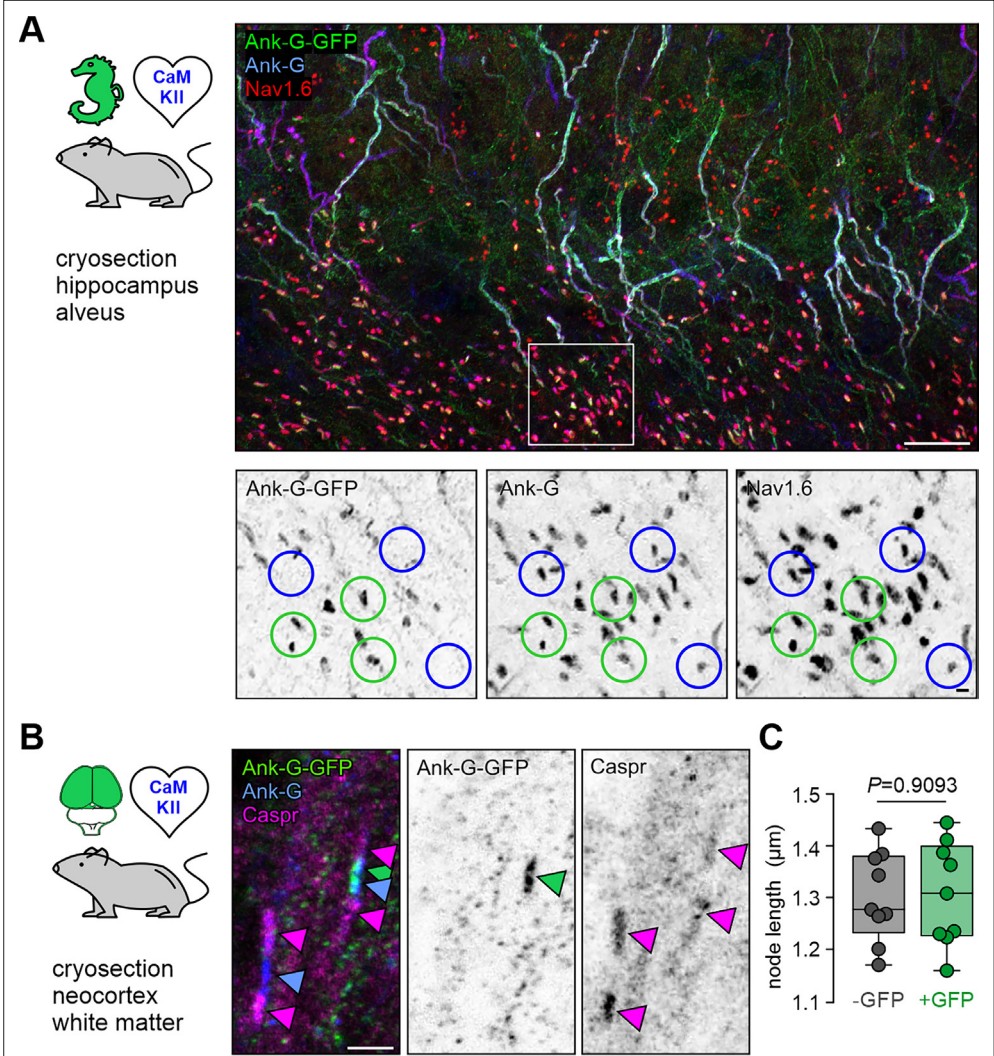

**Figure 3.** Ank-G-GFP activation and expression in nodes of Ranvier do not alter node morphology. (**A**) Cryosection of hippocampal CA3 alveus from an ank-G-GFP × CaMKIIa-Cre mouse, highlighting ank-G-GFP+-node of Ranvier (noR, green circles) and ank-G-GFP–, but ankyrin-G + and Nav1.6+-noR (blue circles) in excitatory neurons. Magnification of the region demarked by a white box is shown in inverted black & white panels. All noR express ankyrin-G (middle) and Nav1.6 (left), but only those belonging to CaMKII + neurons express the ank-G-GFP construct (right). (**B**) Cryosection of neocortical white matter from an ank-G-GFP × CaMKIIa-Cre mouse, highlighting ank-G-GFP+-noR (green arrowhead). Ankyrin-G immunoreactivity (blue arrowheads) is seen in both nodes in the image. Caspr is expressed in paranodal regions of both noR (magenta arrowheads). (**C**) Quantification of the length of noR using the ankyrin-G signal in control (gray) and ank-G-GFP+ neurons (green) in cortical white matter of ank-G-GFP × CaMKIIa-Cre mice shows no difference between the groups (unpaired t-test, n=255 nodes in nine images from three animals). Scale bars A=20 μm, panels in A=2 μm; B=2 μm.

The online version of this article includes the following figure supplement(s) for figure 3:

**Figure supplement 1.** Ank-G-GFP activation and expression in nodes of Ranvier do not alter node morphology.

CaMKII-Cre mice revealed labeling of the CO within the AIS of ank-G-GFP+-neurons (*Figure 5A*) with the typical synpo+ spine apparatus as part of the dendritic domain visible in all samples (*Figure 5A*, magenta puncta in the background). Three-dimensional reconstruction of ank-G-GFP+ AIS indicated numerous synpo clusters residing within the axonal compartment as previously reported (*Schlüter et al., 2017*; *Schlüter et al., 2019*; *Bas Orth et al., 2007*). To test whether GFP-expression had an impact on the normal distribution of synpo clusters in pyramidal neurons, we quantified cluster number per AIS and compared GFP- and GFP+ AIS from the same ank-G-GFP x CaMKII-Cre mice. No

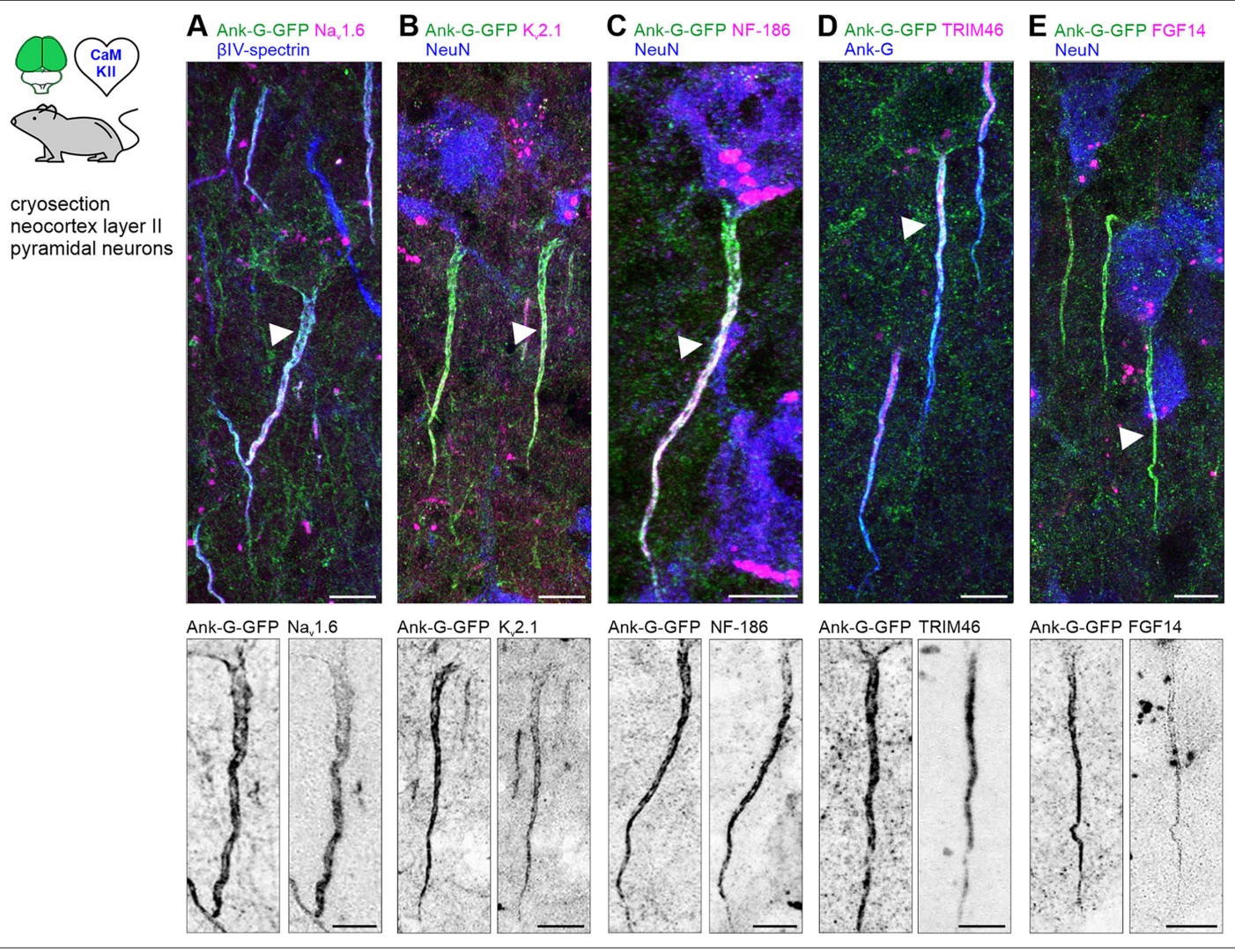

**Figure 4.** Ank-G-GFP expression does not affect the molecular axon initial segment (AIS) composition. (**A**) Merged image of layer II/III pyramidal neurons in S1 of an ank-G-GFP × CaMKII-Cre mouse with intrinsic ank-G-GFP (green), co-labeled against βIV-spectrin (blue) and Nav1.6 (magenta). Bottom: Inverted black & white image of the ank-G-GFP (left) and Nav1.6 (right) signal in a single axon initial segment (AIS) (arrow in merged image). (**B**) Merged image of layer II/III pyramidal neurons in S1 of an ank-G-GFP x CaMKII-Cre mouse with intrinsic ank-G-GFP (green), co-labeled against Kv2.1 (magenta) and NeuN (blue). Bottom: Inverted black & white image of the ank-G-GFP (left) and Kv2.1 (right) signal in a single AIS (arrow in merged image). (**C**) Merged image of a layer V pyramidal neuron in S1 of an ank-G-GFP × CaMKII-Cre mouse with intrinsic ank-G-GFP (green), co-labeled against neurofascin-186 (magenta) and NeuN (blue). Bottom: Inverted black & white image of the ank-G-GFP (left) and NF-186 (right) signal in a single AIS (arrow in merged image). (**D**) Merged image of layer II/III pyramidal neurons in S1 of an ank-G-GFP × CaMKII-Cre mouse with intrinsic ank-G-GFP (green), co-labeled against TRIM46 (magenta) and ankyrin-G (blue). Bottom: Inverted black & white image of the ank-G-GFP (left) and TRIM46 (right) signals in a single AIS (arrow in merged image). (**E**) Merged image of layer II/III pyramidal neurons in S1 of an ank-G-GFP × CaMKII-Cre mouse with intrinsic ank-G-GFP (green), co-labeled against FGF14 (magenta) and NeuN (blue). Bottom: Inverted black & white image of the ank-G-GFP (left) and FGF14 (right) signal in a single AIS (arrow in merged image). All scale bars = 10 μm.

The online version of this article includes the following figure supplement(s) for figure 4:

**Figure supplement 1.** Axon initial segment (AIS) length, position, and molecular composition remains intact after ankyrin-G-GFP expression.

significant difference between GFP⁻ and GFP⁺ AIS was observed (GFP⁻ AIS 3.1±1.7 clusters vs GFP⁺ 3.3±1.7 clusters; n=50 AIS from three animals, p=0.6444, unpaired *t*-test; *Figure 5A*).

Another hallmark of the AIS is its axo-axonic innervation pattern, which we tested using immunostaining against ank-G-GFP, the vesicular GABA transporter (vGAT), a reliable presynaptic marker (*Bragina and Conti, 2018*), and βIV-spectrin (*Figure 5B*). Data revealed that indeed, ank-G-GFP⁺-AIS were contacted by vGAT⁺ axo-axonic interneurons (*Figure 5B*). Conceivably, GFP-expression in

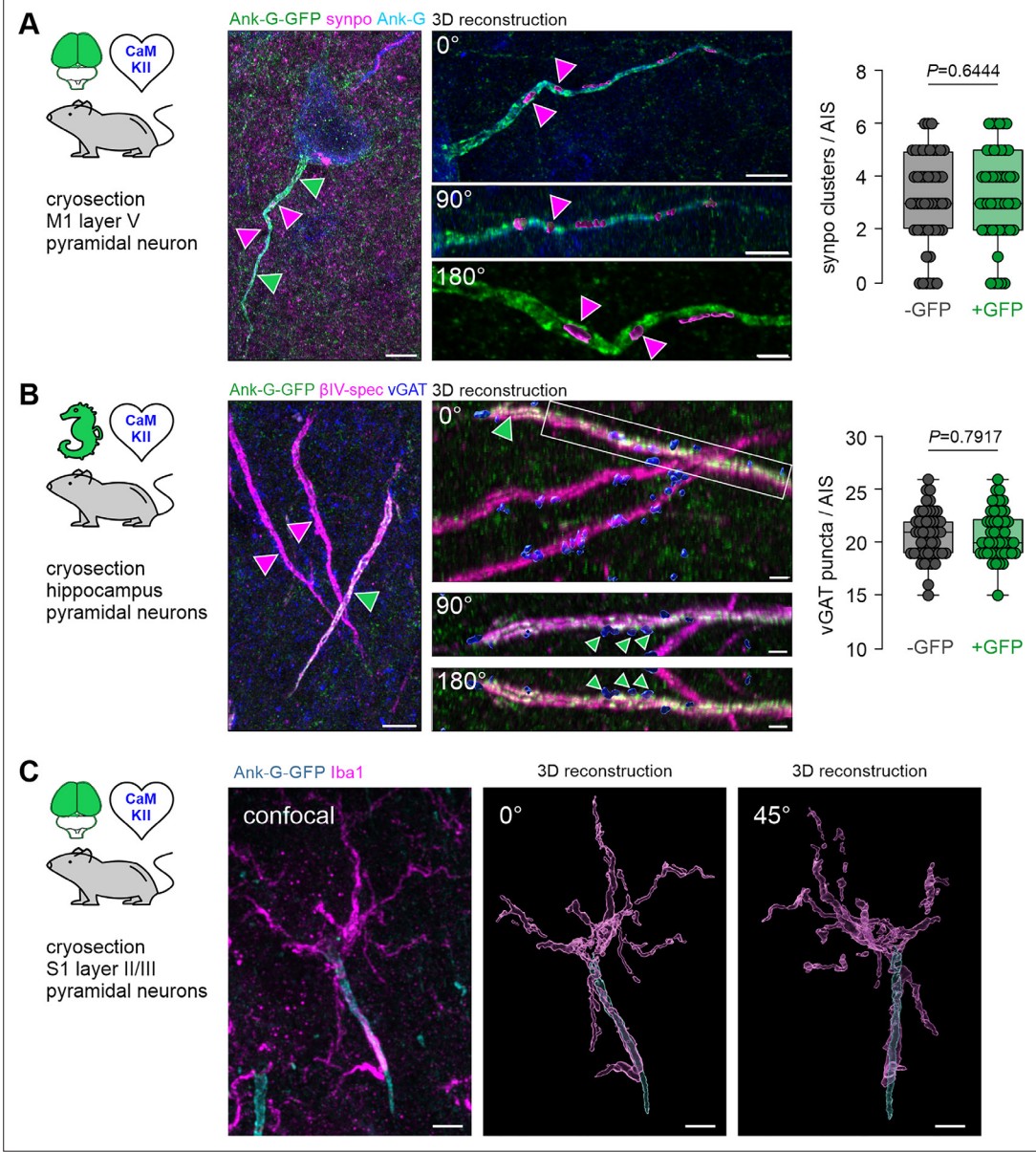

**Figure 5.** Ank-G-GFP expression preserves axonal characteristics. (**A**) Left: Representative image of a single layer V pyramidal neuron in M1 of an ank-G-GFP × CaMKII-Cre mouse with intrinsic ank-G-GFP (green), and colabeling against ankyrin-G (blue) and synaptopodin (synpo: magenta, arrows). Right: 3D-reconstruction of the axon initial segment (AIS) shown on the left. Three different rotations (0, 90, and 180°) of the same AIS indicate synpo clusters within the confinement of the axonal membrane of the AIS (magenta arrowheads). Scale bar left = 5 µm, right upper and middle panel = 5 µm, bottom panel = 2 µm. Quantification of synpo cluster number in AIS derived from ank-G-GFP × CaMKII-Cre mice showed no significant difference between GFP- and GFP +AIS (n=50 AIS from three animals, p=0.6444, unpaired t-test, whiskers from min to max values). (**B**) Left: Representative image of three AIS of pyramidal neurons in CA1 of an ank-G-GFP × CaMKII-Cre mouse with intrinsic ank-G-GFP (green), and colabeling against vGAT (blue) and βIV-spectrin (magenta). The only ank-G-GFP+AIS in this image is highlighted by a green arrowhead; ank-G-GFP- AIS are demarked by magenta arrowheads. Right: 3D-reconstruction of the GFP +AIS shown on the left. Three different rotations (0, 90, and 180°) indicate vGAT clusters (green arrowheads) along the AIS where GABAergic synapses innervate the axon. Note that both ank-G-GFP+, as well as ank-G-GFP- AIS are contacted by vGAT puncta (blue). Scale bar left = 10 µm, right upper and middle panel = 5 µm, bottom panel = 2 µm. Quantification of vGAT + puncta along AIS derived from ank-G-GFP × CaMKII-Cre mice showed no significant difference between GFP- and GFP +AIS (n=50 AIS from three animals, p=0.7917, unpaired t-test, whiskers from min to max values). (**C**) Left: Representative image of two AIS of layer II/III pyramidal neurons in S1

*Figure 5 continued on next page*

*Figure 5 continued*
of an ank-G-GFP × CaMKII-Cre mouse with intrinsic ank-G-GFP (green), and colabeling against Iba1 (magenta), a marker for microglia. Right: 3D-reconstruction of the boxed region in the left panel. Two different rotations (0 and 45°) indicate that this AIS is contacted by at least one microglial process. Scale bar = 5 μm.

ank-G-GFP mice could have an impact on the number of putative GABAergic synapses at the AIS. Therefore, we quantified the number of vGAT⁺ puncta along GFP⁻ and GFP⁺ AIS from the same ank-G-GFP x CaMKII-Cre mice. We found no significant difference between these groups (GFP⁻ AIS 20.7±2.3 vGAT puncta vs GFP⁺ 20.8±2.2 vGAT puncta; n=50 AIS from three animals, p=0.7917, unpaired *t*-test; *Figure 5B*). We note that GFP⁻ and GFP⁺ AIS belong to genetically different cell types (defined by CamKIIa expression) that might not necessarily feature the same synpo or vGAT distributions.

A further characteristic of the AIS is its occasional ensheathing in microglial processes, which is the case for approximately 4% of all AIS in rodent cortex (*Baalman et al., 2015*). We, therefore, performed immunostaining against the microglial marker Iba1 in ank-G-GFP × CaMKII mice and found scarce, but clearly distinguishable interaction of select GFP⁺ AIS with microglial processes in S1 layer II/III pyramidal neurons (*Figure 5C*). In those cases, the majority of the microglial cell body was in contact with the soma of a pyramidal neuron and extended one or more processes alongside the AIS towards its distal end (*Figure 5C*).

## The AIS nanostructure is maintained after GFP fusion to ankyrin-G

The nanostructure of the AIS scaffold in cortical neurons exhibits a stereotypical subcellular organization, in which a periodic spacing of actin and spectrin rings intercalates with voltage-gated ion channels at approximately 170–190 nm distance, depending on the cell type studied (*Leterrier et al., 2015*; *Schlüter et al., 2019*; *Akter et al., 2020*). Both concentration and spatial arrangement of ion channels are thought to be crucial for AP generation and modulation (*Lazarov et al., 2018*; *Hu et al., 2009*; *Inda et al., 2006*; *Kress et al., 2010*) and, therefore, constitute a hallmark of excitable neuronal microdomains such as the AIS.

Here, we aimed to verify that the introduction of GFP into this scaffold did not disturb its spatial organization. Dissociated hippocampal neurons derived from the ank-G-GFP line without prior Cre exposure were infected with lentiviruses expressing functional Cre-recombinase (+Cre) or a deficient version of Cre without recombinase activity as control (ΔCre), both fused to nuclear GFP (nGFP) (*Figure 6*). Conventional confocal microscopy after immunostaining against the AIS marker βIV-spectrin showed that ΔCre neurons exhibited no GFP staining at their βIV-spectrin⁺ AIS (*Figure 6A* left), whereas intrinsic GFP labeling of the AIS was achieved in cultures exposed to functional Cre-recombinase (*Figure 6A* middle). Using Stimulated Emission Depletion (STED) microscopy, we assessed the nanoscale distribution of ankyrin-G, both via intrinsic GFP label as detected by the FluoTag-X4-GFP nanobody (*Figure 6A* right, upper panel) as well as via antibody staining directed against the C-terminus of ankyrin-G (*Figure 6A* right, lower panel). To further elucidate the nanoscale distribution of AIS proteins in control vs Cre-recombinase exposed cells, we performed immunostaining against the C-terminus of ankyrin-G (*Figure 6B*, left panel), βIV-spectrin (*Figure 6B*, middle panel), and the voltage-gated potassium channel K$_v$1.2 (*Figure 6B*, right panel), which has been shown to cluster at the AIS (*Lorincz and Nusser, 2008*). Comparing ΔCre to +Cre neurons, we found an equally reproducible staining for all three markers (compare upper and lower rows in *Figure 6B*). Autocorrelation analysis verified the characteristic 190 nm periodic organization of the proteins investigated (*Figure 6C*; left = ankyrin G, middle = βIV-spectrin, right = K$_v$1.2). Further, an autocorrelation amplitude analysis, which calculates the difference between the autocorrelation value at 190 nm and the average values at 95 nm and 285 nm, respectively, was performed, showing no significant difference between untreated, ΔCre or +Cre ank-G-GFP cultures (*Figure 6D*; left = ankyrin G, middle = βIV-spectrin, right = K$_v$1.2). In summary, our data indicate that ank-G-GFP⁺-AIS develop normally and upon Cre-activation, retain their nanoscale organization.

## Cellular excitability is not affected by GFP fusion to ankyrin-G

In order to utilize the ank-G-GFP line to study electrophysiological consequences associated with AIS function and plasticity, we verified that the fusion protein itself does not change the basic excitatory

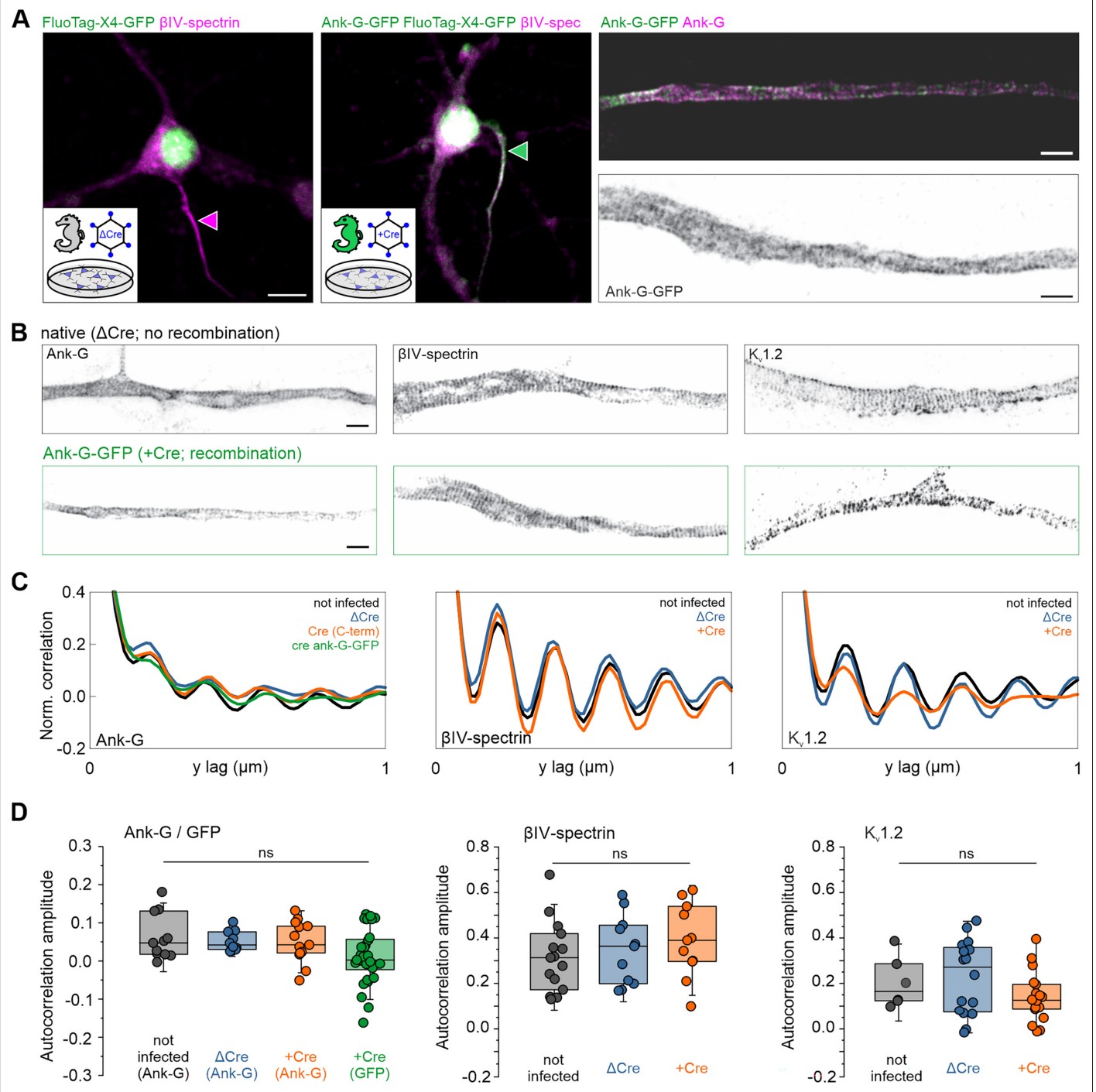

**Figure 6.** The axon initial segment (AIS) nanostructure is maintained after GFP fusion to ankyrin-G. (**A**) Left: Representative fluorescence image of dissociated hippocampal neurons in vitro, transduced either with lentivirus expressing recombination-deficient ΔCre fused to nuclear GFP (magenta arrowhead) or with Cre-recombinase fused to nGFP (green arrowhead). Right: STED images of AIS labeled with intrinsic GFP detected by the FluoTag-X4-GFP nanobody (green) and by an antibody against the C-terminus of ankyrin-G (magenta). Scale bars left and middle = 10 µm, right = 1 µm. Note, the nuclear GFP signal is derived from the virus and not the ank-G-GFP label. (**B**) Representative STED images of the nanoscale organization of the C-terminus of ankyrin-G (left), βVI-spectrin (middle), and Kv1.2 (right) in neurons infected with either ΔCre (top) or Cre virus (bottom panels). AIS shown in (**B**) (left bottom panel) is the same as the dual color panel A (right panel). Scale bars = 1 µm. (**C**) Autocorrelation analysis shows the characteristic ~190 nm periodic organization of the proteins imaged in (**B**). Note how the periodicity of the ank-G-GFP matches the periodicity of the C-terminus of ankyrin-G in all conditions. (**D**) Autocorrelation amplitude analysis of ankyrin-G and ankyrin-G-GFP (left panel), βIV-spectrin (middle panel), and Kv1.2 (right panel) is not statistically different in cultures treated with ΔCre, +Cre, or untreated (one-way ANOVA). Ank-G-GFP data (green) indicates the signal of the GFP label itself. For +Cre conditions, only GFP +AIS were used for analysis.

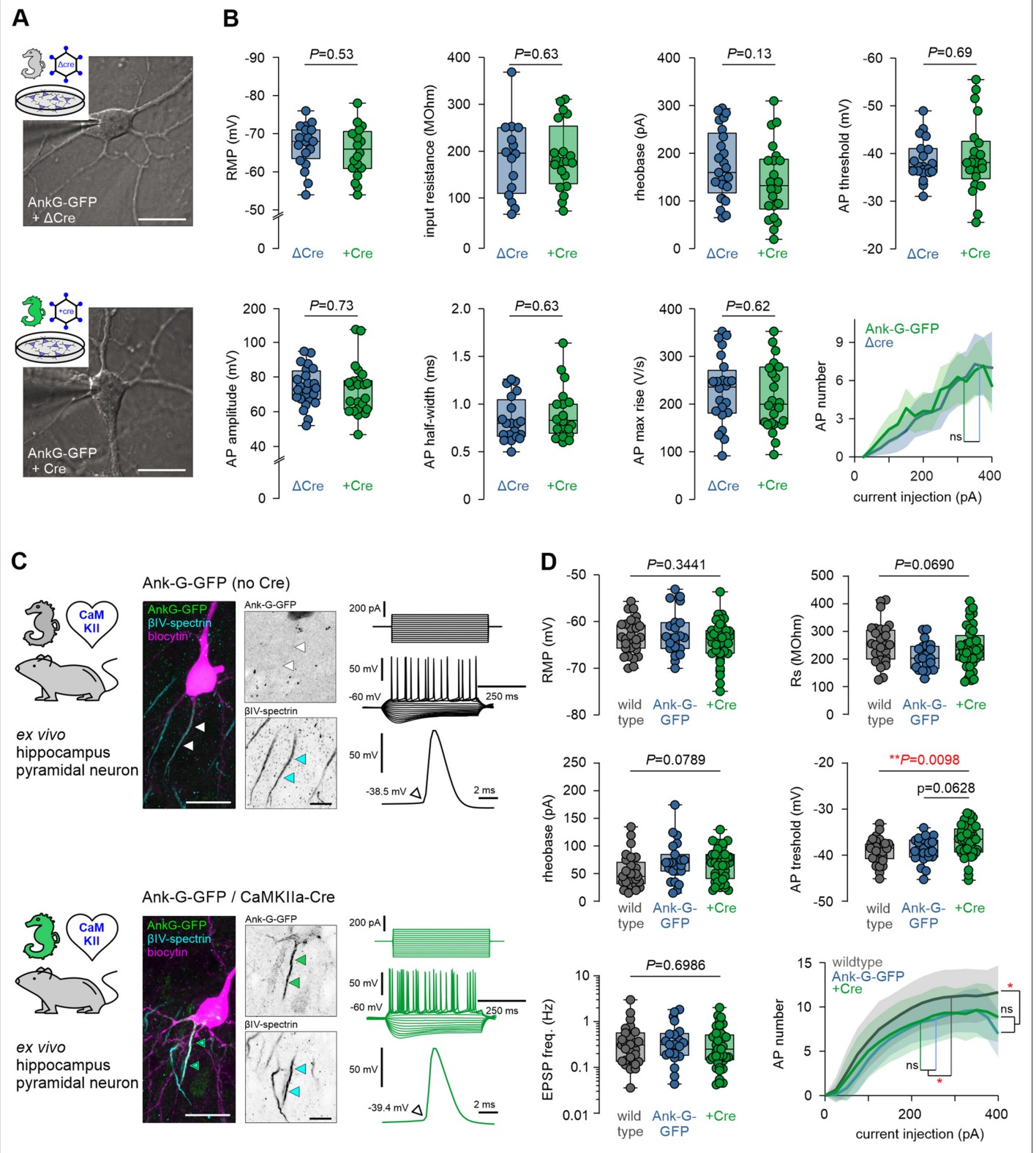

**Figure 7.** Neuronal excitability is not affected by GFP fusion to ankyrin-G. (**A**) Representative images of a ΔCre control neuron (upper panel) and a+Cre neuron (lower panel) from dissociated hippocampal cultures with the patch pipette attached. Scale bar = 25 μm. (**B**) Analysis of active and passive electrophysiological membrane properties, all indicating no significant changes between experimental and control groups (One-way ANOVA, Holm-Sidak's multiple comparison test. p-values are indicated in each graph, n=25 ΔCre cells, 24+Cre cells). (**C**) Representative images of a biocytin-filled (magenta) hippocampal pyramidal neuron from an ank-G-GFP control (upper panel) and an ank-G-GFP × CaMKII-Cre animal (lower panel), colabeled for

*Figure 7 continued on next page*

*Figure 7 continued*

βIV-spectrin (blue) with intrinsic GFP signal (green). The black & white panels highlight the individual axon initial segment (AIS) in each cell. Note that in the control neuron (upper), no GFP signal for the AIS could be detected (white arrowheads). Scale bar = 20 µm. Representative traces of action potential (AP) trains elicited by current injection (250 ms, –200 to +250 pA) in control (upper panel) and ank-G-GFP × CamKII-Cre neurons (lower panel, green). (**D**) Analysis of RMP, Rs, AP rising phase, AP threshold, rheobase, and maximum AP number, all indicating no significant changes between experimental and control groups (One-way ANOVA and Tukey's multiple comparisons test, or Kruskal-Wallis and Dunn's multiple comparison test (non-parametric). P-values are indicated in each graph, n=28 Ctrl cells, 23 ank-G-GFP cells, 42 ank-G-GFP × CamKII-Cre cells. Whiskers in figure show min to max of values).

properties of ank-G-GFP$^+$ neurons in different systems. Therefore, we investigated whether cells expressing the fusion protein exhibit different electrophysiological properties as compared to wild-type cells. We used two common model systems: (i) dissociated hippocampal neurons infected either with Cre recombinase or a recombination-deficient ΔCre virus (***Figure 7A and B***), and (ii) hippocampal neurons in acute slices of adult mice that were generated from the ank-G-GFP x CaMKII-Cre matings. These were compared to acute slices from both ank-G-GFP littermate controls not exposed to Cre, and standard wild-type C57Bl/6 J mice (***Figure 7C and D***).

In dissociated hippocampal neurons, we first determined passive membrane properties such as resting membrane potential (RMP) and input resistance ($R_s$) and found no significant difference between neurons expressing Cre-recombinase and ΔCre controls (***Figure 7B***). Similarly, AP properties, including amplitude, half-width, voltage threshold, maximum rise time, and rheobase were unaffected (***Figure 7B***). Plotting the input/output relation with a comparison of the maximum slope f' (max) of the I/F-curve further provided evidence that the GFP fusion construct does not alter the firing properties of infected neurons in vitro.

The same electrophysiological parameters were tested in ex vivo hippocampal acute slices from three groups: (i) wild-type mice, (ii) ank-G-GFP mice without Cre exposure, and (iii) ank-G-GFP x CamKII-Cre mice (***Figure 7C and D***). We found no evidence of an impact of the GFP fusion protein on any of the investigated passive and active properties (***Figure 7C and D***, ***Supplementary file 1A***).

## Ank-G-GFP$^+$ AIS undergo structural plasticity after changes in network state in vitro and ex vivo

Previous post-hoc studies of AIS plasticity indicated that the AIS scales its length, and sometimes position, in response to changes in network activity (reviewed in ***Jamann et al., 2018***). Using the new live reporter, we tested whether the same principles apply to ank-G-GFP$^+$-AIS. We employed previously reported experimental conditions of network alterations ***Jamann et al., 2021***; ***Grubb and Burrone, 2010***; ***Engelhardt et al., 2018*** to elicit structural AIS changes ex vivo (***Figure 8A***). To that end, hippocampal OTC were prepared from ank-G-GFP mice at P4 and transfected with an AAV pmSyn1- EBFP-Cre virus at DIV 1 to trigger GFP expression.

Since previous research indicated that AIS elongation is a process that requires a longer time frame of network silencing (days to weeks; ***Jamann et al., 2021***; ***Gutzmann et al., 2014***), we first blocked synaptic transmission via NMDA receptors by maintaining OTC in the presence of 10 mM MgSO$_4$ for 20 d. Under these conditions, AIS elongated significantly (Ctrl 27.9±3.0 µm vs MgSO$_4$ 32.1±2.4 µm, n=6 OTC, 50 AIS per OTC; ***Figure 8A and B***). The same effect was observed when cultures were maintained in MgSO$_4$ for 10 d (Ctrl 32.1±3.5 µm vs MgSO$_4$ 35.9±1.2 µm, n=6 OTC, 50 AIS per OTC; ***Figure 8C***). Induction of increased network activity by exposing OTC to chronic depolarizing conditions (6 mM KCl) for 20 d and 10 d, respectively, did not result in significant length changes (KCl$_{(10 d)}$ 29.9±1.8 µm, KCl$_{(20 d)}$ 27.0±2.7 µm, n=6 OTC, 50 AIS per OTC; ***Figure 8B and D***, one-way ANOVA and Tukey's test for B, unpaired t-tests for C and D).

AIS plasticity is proposed as a homeostatic mechanism (***Wefelmeyer et al., 2016***; ***Jamann et al., 2021***) and should be reversible as network activity returns to baseline. We, therefore, tested the reversibility of AIS plasticity by applying rescue conditions. After initial exposure to the activity-altering conditions for 10 d, OTC were cultured under normal growth conditions for an additional 10 d (to DIV 20; ***Figure 8A***). In MgSO$_4$-treated OCT, AIS length returned to baseline after rescue (Ctrl MgSO$_4$ 27.9±3.0 µm vs rescue 30.6±2.3 µm; n=6 OTC, 50 AIS per OTC; ***Figure 8C***). Rescue paradigms in KCl-treated cultures did not affect AIS length as expected (Ctrl KCl 27.9±3.0 µm vs rescue 28.9±1.9 µm; n=6 OTC, 50 AIS per OTC; ***Figure 8D***, all tests performed as unpaired *t*-test).

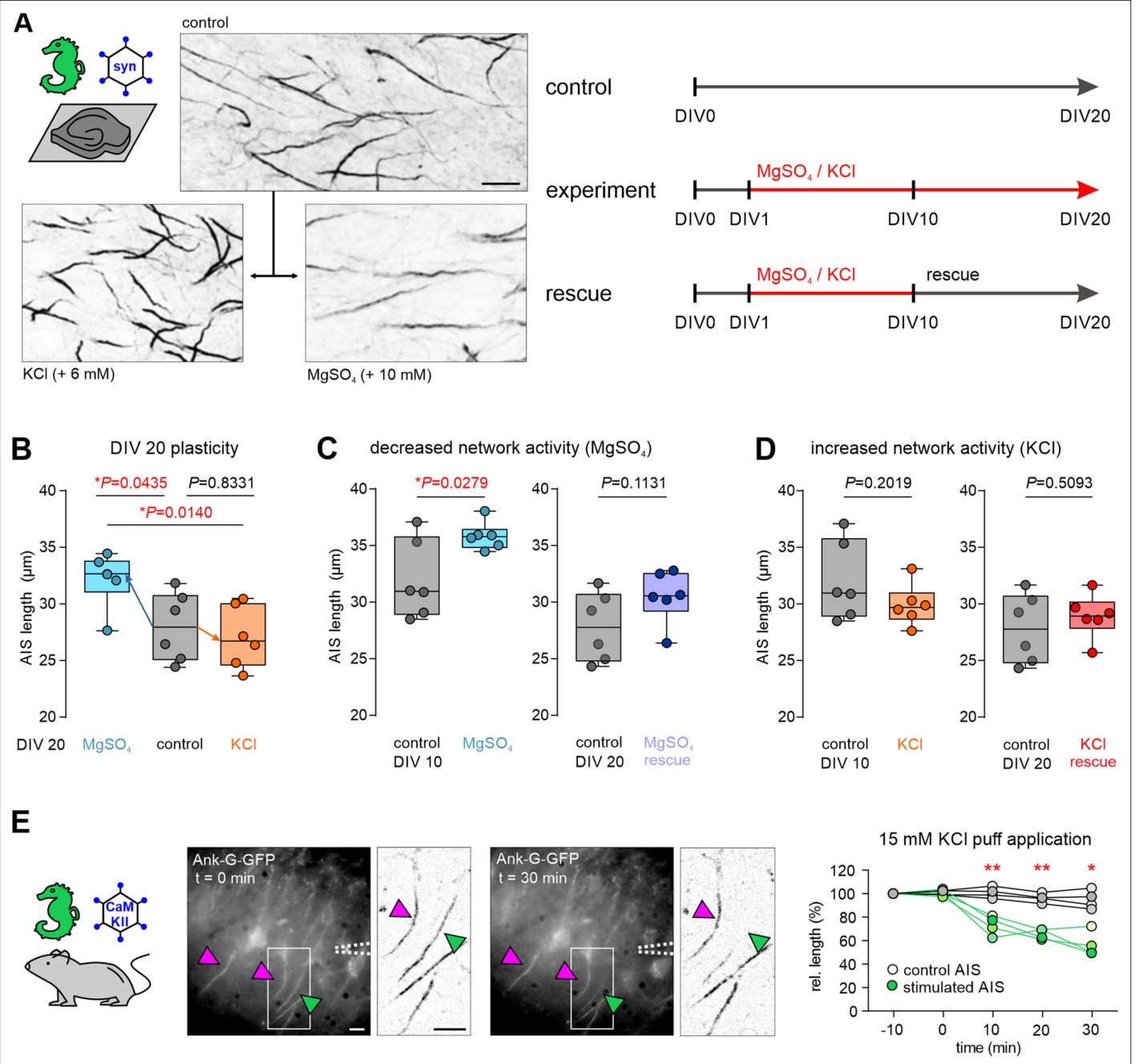

**Figure 8.** Ank-G-GFP+AIS exhibit structural plasticity in conditions of altered network activity in vitro. (**A**) Representative images of ank-G-GFP labeled axon initial segment (AIS) in organotypic cultures (OTCs). Lower left panel: AIS after exposure to 6 mM KCl. Lower right panel: AIS after exposure to 10 mM MgSO4, both for 20 DIV. Scale bar = 10 μm. Experimental procedures are outlined in the cartoon, indicating treatment duration and DIV. (**B**) Left in graph: AIS elongation after exposure of OTC to 10 mM MgSO4 for 20 d. Mean AIS length was increased by 4.2 μm compared to controls (n=6 OTC, 50 AIS per OTC). Right in graph: AIS remain unchanged after exposure of OTC to 6 mM KCl for 20 d (n=6 OTC, 50 AIS per OTC, one-way ANOVA and Tukey's test). (**C**) Under conditions of decreased network activity (10 mM MgSO4 for 10 d), AIS are elongated (unpaired t-test, n=6 OTC, 50 AIS per OTC). Rescue conditions (normal growth medium) from DIV 10–20 resulted in a return to baseline AIS length (unpaired t-test, n=6 OTC, 50 AIS per OTC). (**D**) Under conditions of increased network activity (6 mM KCl for 10 d), no changes in AIS length were observed (unpaired t-test, n=6 OTC, 50 AIS per OTC). Consequently, no AIS length changes were seen after applying rescue conditions from DIV 10–20 (normal growth medium; unpaired t-test, n=6 OTC, 50 AIS per OTC). (**E**) Rapid AIS shortening in hippocampal acute slices. The GFP construct was activated by injection of an AAV5-CaMKIIa-Cre virus and acute slices were prepared and maintained in ACSF. KCl (15 mM) was applied to a selected cell body, which showed a rapid decline of GFP fluorescence along the AIS within 10 min of the application (green arrowhead also in inverted images). Neurons that were outside of the KCl application showed no significant length changes (magenta arrowheads, also in inverted images). Change in the relative length of individual AIS targeted by high

*Figure 8 continued on next page*

*Figure 8 continued*

potassium (green) and surrounding AIS (gray) is plotted against the duration of the application (four pairs of sample and control AIS, t-tests, scale bar = 10 µm. Whiskers in figure show min to max values).

AIS shortening after increased network activity occurs in a rapid time frame (1 hour) and is bidirectional with a return to baseline after an initial length change (*Jamann et al., 2021*). To test whether rapid AIS plasticity could be observed live in ank-G-GFP mice, we applied a different strategy. Ank-G-GFP mice were injected with an AAV5-CaMKIIa-Cre virus triggering Cre expression in a subset of hippocampal pyramidal neurons (*Figure 8E*). After 3 wk of virus expression, stable ank-G-GFP signals were detected in a subset of pyramidal neurons using epifluorescence in acute slices. Using pipette application of 15 mM KCl targeting individual neurons, we observed rapid changes in fluorescent signals based on the intrinsic GFP (*Figure 8E*). Direct application of high potassium resulted in a relative length reduction only of the AIS associated with the affected neuron, while all surrounding labeled AIS maintained their initial length (n=4 pairs of sample and control AIS; 10 min Ctrl vs KCl: p=0.0068; 20 min Ctrl vs KCl: p=0.0045; 30 min Ctrl vs KCl: p=0.0130; paired *t*-tests, *Figure 8E*). Interestingly, within minutes of the KCl application to the axon hillock, we observed the partial disappearance of the proximal onset of an individual AIS (*Figure 8E*, insets). This was quite striking since the majority of previous studies on this topic could only access AIS length and position changes in a post-hoc manner and never in the same cell. Using patch-clamp recordings, we verified that neurons tolerate and survive the treatment in the respective time frame. We here demonstrate that actual structural AIS changes do occur rapidly within 10 min and that they can be visualized and monitored within the same neuron.

## Live imaging of ank-G-GFP+-AIS in vivo reveals long-term stability of AIS length

As outlined above, previous studies investigating AIS plasticity relied on post-hoc analysis in fixed sections. We conducted the ultimate test of the ank-G-GFP reporter using live cell imaging in intact animals in a longitudinal approach.

Three different in vivo systems were applied to test the reliability, reproducibility, and stability of intrinsic ank-G-GFP signals in vivo. First, we imaged and tracked the same pyramidal neuron AIS in S1 for 1 wk (*Figure 9A–C*). After implanting cranial windows in 3 mo-old ank-G-GFP × CaMKII-Cre animals, the AIS of layer II/III neurons in S1 were imaged based on their intrinsic GFP signal, which proved to be robust and easy to visualize (*Figure 9A–C*). We compared the same region of interest at a 7 d interval and analyzed the length of the same individual AIS (*Figure 9B and C*). Quantification revealed that the same AIS maintained their average length reliably (week 1, magenta: 17.3±3.9 µm vs week 2, green: 16.8±3.9 µm, paired *t*-test, n=30 AIS; *Figure 9B*). This indicated that neither previous imaging nor potential physiological alterations in a 1 wk time frame changed GFP localization in such a way that it caused global length differences. Plotting of mean AIS length from week 1 and week 2 indicated a strong and significant correlation among individual AIS lengths measured in the same ROI ($r^2$=0.7892, p<0.0001, n=30 AIS; *Figure 9B*).

In addition to constitutive expression of ank-G-GFP in intercrossed transgenic mice, AIS could also be monitored in vivo using a viral approach in two additional systems: (1) specific subpopulations of cross-callosal pyramidal neurons of the caudal forelimb region of M1 (*Figure 9D*) and (2) principal neurons of deep brain areas, like the basolateral amygdala (BLA) via gradient reflective index (GRIN) lenses (*Figure 9E*).

Our live imaging data provides evidence that the ank-G-GFP model is a powerful tool to study excitable membrane domains in living animals in future studies of neural development, animal behavior, and disease progression.

## Discussion

Here, we introduce a mouse line that allows Cre-dependent expression of an ank-G-GFP fusion protein using the native *Ank3* locus, thus enabling the monitoring and analysis of AIS and noR parameters in living mice. Our data show (i) efficient, reproducible labeling of the AIS and noR in genetically defined neuronal populations and with temporal precision in vitro, ex vivo, and in vivo, (ii) normal AIS

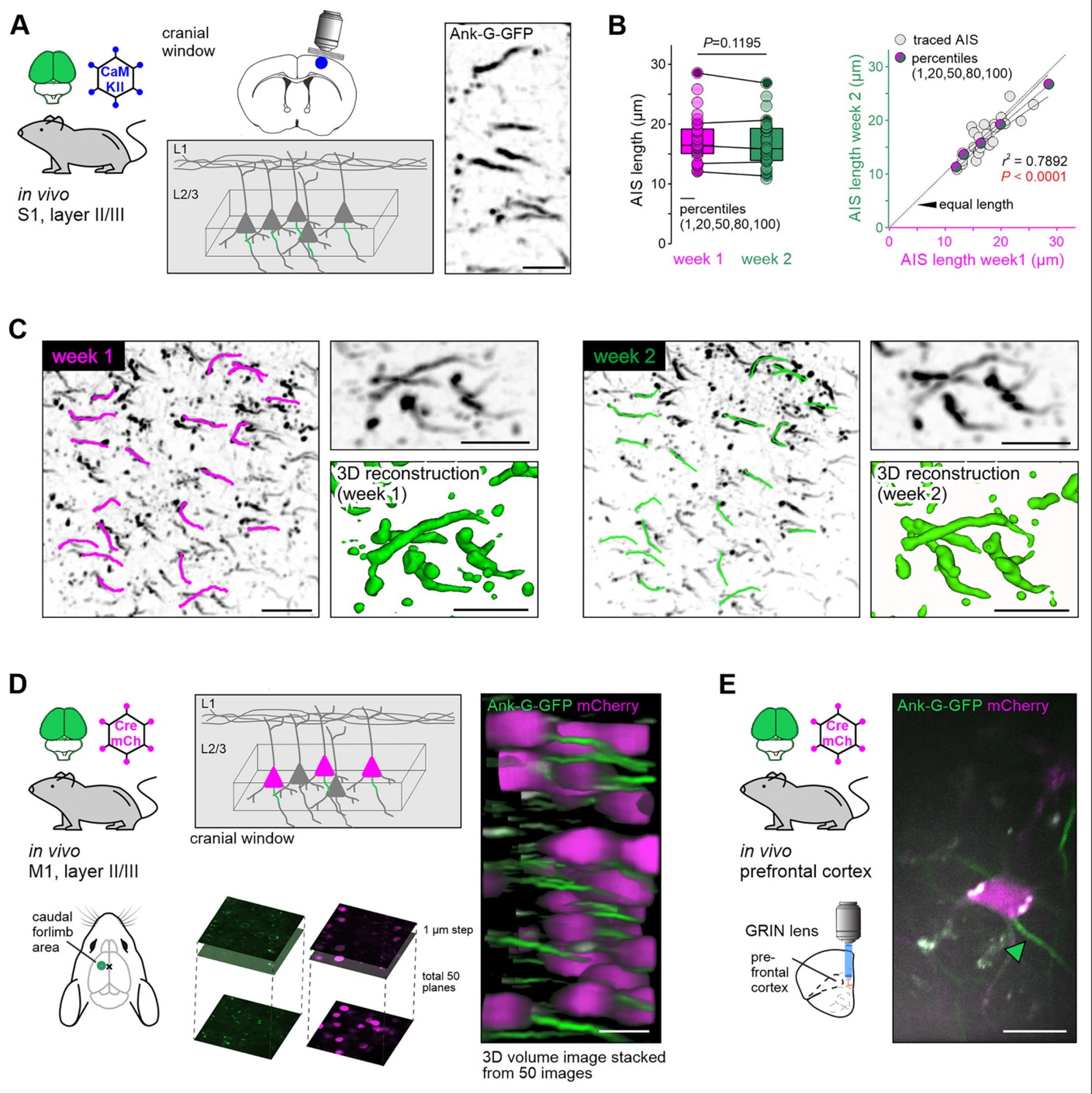

**Figure 9.** Live imaging of ank-G-GFP+-AIS in vivo. Experimental setup: (**A**) cranial window was implanted in ank-G-GFP × CaMKII-Cre animals (two animals, 3–4-mo-old). Imaging supragranular neurons in S1 layer II/III allowed for stable GFP-signal visualization (right panel, inverted GFP signal). Scale bar = 10 μm. (**B**) AIS length analysis from the same individual axon initial segment (AIS) 1 wk apart showed no significant differences between the time points and ROI (paired t-test, p=0.1195; 30 AIS). Linear regression analysis indicated a strong correlation between individual AIS length in the same neurons from week 1 (magenta) and week 2 (green; r2=0.7892, p<0.0001, n=30 AIS from one ROI and animal). (**C**) 2PM revealed a robust and stable GFP signal during the duration of the imaging session. Individual AIS were easily detectable and maintained their overall geometry over the chosen time course (insets with 3D reconstruction). Scale bar overview = 20 μm, close up = 10 μm. Left ROI from week 1 imaging session. Magenta traces along individual AIS demark those subjected to length measurements. Right The same ROI as in the left panel a week later, with the same AIS marked for length analysis (green). Scale bar = 20 μm. (**D**) Experimental setup: Ank-G-GFP animals were injected contralateral to the imaging site with a retrograde Cre-recombinase and mCherry-expressing AAV resulting in GFP expression at the AIS of mCherry-positive neurons in M1 (caudal forelimb

*Figure 9 continued on next page*

*Figure 9 continued*

area). Imaging via a cranial window revealed robust GFP signal in infected cells. 3D stacks were produced 150–200 μm from the dura mater; stacks were imaged with 1 μm intervals and 50 images merged into a volume. Scale bar = 20 μm. (**E**) Experimental setup: Ank-G-GFP animals were injected with a retrograde Cre-recombinase and mCherry-expressing AAV in the medial prefrontal cortex resulting in GFP expression at the AIS of mCherry-positive neurons in the basolateral amygdala. Imaging was performed through a GRIN lens and allowed for visualization of individual mCherry-positive neurons and their respective AIS (green arrowhead). Scale bar = 10 μm.

properties, including clustering of binding partners, inhibitory synaptic clustering, and nanostructure in ank-G-GFP mice, (iii) normal electrophysiological properties, including AP properties and excitatory synaptic function in ank-G-GFP mice, (iv) structural AIS plasticity ex vivo after pharmacological induction of network changes, and importantly (v) reliable tracking of ank-G-GFP-labeled AIS over days and weeks in vivo.

An increasing number of studies have shown that neurons respond to changes in neuronal activity by altering the length, position, and/or channel distribution of the AIS (*Wefelmeyer et al., 2016*; *Jamann et al., 2021*; *Grubb and Burrone, 2010*; *Kuba et al., 2010*; *Zbili et al., 2021*; *Yamada and Kuba, 2016*). For example, sensory deprivation elicits lengthening of the AIS and a corresponding increase in neuronal excitability in the barrel cortex (*Jamann et al., 2021*). Conditions that mimic mild traumatic brain injury cause decreases in AIS length in the somatosensory cortex (*Vascak et al., 2017*). AIS plasticity in response to changes in network activity has further been observed in the avian (*Kuba et al., 2010*) and rodent auditory system (*Kim et al., 2019*), isolated hippocampal neurons (*Grubb and Burrone, 2010*), hippocampal OTC (*Wefelmeyer et al., 2015*), and numerous other models (reviewed in *Jamann et al., 2018*). However, a limitation of these studies is the post-hoc analysis of neuronal populations from different animals, which confines our understanding of the temporal regulation of this process at the single-cell level. The ank-G-GFP model allows live imaging of the AIS under different physiological and pathophysiological conditions, generating a novel approach to the investigation and discovery of structure/function relationships and the temporal regulation of these processes in health and disease. While significant inroads into the understanding of AIS assembly and plasticity have been made recently (*Fréal et al., 2019*; *Harterink et al., 2019*; *Le Bras et al., 2014*; *Dorrego-Rivas et al., 2022*), this live reporter offers a unique tool to truly understand the actual molecular mechanisms involved.

Ankyrin-G plays a critical role in the establishment of the AIS and is responsible for the clustering of known AIS proteins, including voltage-gated ion channels, cell adhesion molecules, and inhibitory synaptic components (*Jenkins et al., 2015*; *Nelson and Jenkins, 2017*; *Zhou et al., 1998*). Given this requirement for ankyrin-G to assemble the AIS, the use of GFP-tagged native ankyrin-G as a molecular marker for the AIS allows early insight into AIS assembly in vivo. Beyond the initial ankyrin-G-dependent assembly of the AIS, other components of the AIS participate in its stability, including voltage-gated sodium channels, βIV-spectrin, and neurofascin (*Leterrier et al., 2017*; *Lacas-Gervais et al., 2004*). Therefore, it is crucial to understand the temporal and spatial regulation of these interactions under normal steady-state conditions and during events of plasticity.

Some components of the AIS, including voltage-gated sodium channels, alter their position along with ankyrin-G during homeostatic plasticity (*Grubb and Burrone, 2010*), indicating a tight coupling of ankyrin-G and some binding partners. Other AIS constituents, like ankyrin-G-dependent GABAergic synapses, can uncouple their AIS position from that of ankyrin-G (*Wefelmeyer et al., 2015*). Thus, future experiments with the ank-G-GFP mouse could combine live cell markers for other AIS proteins to study the mechanisms of AIS localization and plasticity and how these proteins are coupled and/or uncoupled from ankyrin-G during the development and maturation of the CNS. A recent study utilizing this mouse line provided new insights into rapid AIS plasticity mediated by the rapid internalization of AIS-localized sodium channels (*Fréal et al., 2023*).

Beyond the AIS, ankyrin-G is also localized to other subcellular locations within neurons. For example, ankyrin-G is highly concentrated at noR, where it participates in the clustering of voltage-gated ion channels at the nodal membrane (*Rasband and Peles, 2021*). Similar to what has been observed with the AIS, post-hoc approaches have identified variability in noR length, which could theoretically result in the regulation of AP propagation velocity (*Arancibia-Cárcamo et al., 2017*). Here, we demonstrate efficient labeling of noR in ank-G-GFP mice, underlining the ank-G-GFP mouse as a useful tool for monitoring size and position as well as glial interaction of noR in vivo. In addition

to noR, ankyrin-G is also localized to the somatic membrane in mature neurons of the neocortex and hippocampus, where it regulates GABAergic synaptic density (*Tseng et al., 2015*). Consistent with our previous results, we observe ank-G-GFP accumulation on somatic and proximal dendritic membranes in hippocampal pyramidal neurons. Finally, the 190 kDa isoform of ankyrin-G localizes to subdomains within dendritic spines, where it plays an important role in glutamate receptor stability and dendritic spine morphology (*Smith et al., 2014*). Since the ank-G-GFP mouse uses a carboxy-terminal GFP tag that would be present on all three major classes of *Ank3* transcripts in the CNS (190, 270, and 480 kDa isoforms), future experiments with the ank-G-GFP line could examine ankyrin-G dynamics within dendritic spines during synaptic plasticity.

In summary, we here provide a novel reporter mouse model for the next generation of experiments focusing on excitable axonal microdomains in living tissue, a so far unattainable goal. AIS plasticity remains a highly relevant, yet understudied, core neuronal mechanism that has a significant impact on how neurons integrate and participate in local networks. The actual mechanisms of AIS plasticity remain largely unknown. Their investigation would greatly benefit from this model, allowing for the visualization of subcellular modifications during diverse network states, including in disease models. The ank-G-GFP reporter mouse, in combination with the high temporal and spatial resolution now obtained by e.g., voltage or calcium imaging, will move an entire research field forward. In addition, the added benefit of also studying noR and other ankyrin-G-dependent structures including those in the peripheral nervous system, will open up avenues of investigations that have so far not been available to researchers.

## Materials and methods

### Generation of mouse line

We used the Flip-excision (FLEx) system (*Schnütgen et al., 2003*) to introduce a conditional GFP-tag into the final exon of ankyrin-G, the major scaffolding protein of the AIS (*Figure 1A*; *Leterrier, 2018*). A FLEx cassette containing the last exon of *Ank3* (ENSMUSE00001313132, exon 42) followed by a copy of the last exon of *Ank3* fused with the coding sequence for eGFP in the reverse orientation was generated using standard molecular methods. The entire cassette was flanked by alternating loxP and lox2272 sites. In the presence of Cre recombinase, the cassette orientation is irreversibly flipped by sequential recombination and excision of two of the lox sites (*Figure 1B*). The result is the expression of an ankyrin-G-GFP fusion product from the endogenous locus only in cells expressing Cre recombinase (*Figure 1B*). The linker sequence between *Ank3* and GFP is the same sequence used in previously described 190-, 270-, and 480-kDa ankyrin-G-GFP plasmids (*Kole and Stuart, 2012*; *Jenkins et al., 2015*; *Kizhatil and Bennett, 2004*; *Zhang and Bennett, 1998*). Importantly, these ankyrin-G-GFP fusion products have been able to rescue all functions of ankyrin-G tested to date, suggesting that the addition of GFP to the ankyrin-G carboxy-terminus does not affect ankyrin function (*Jenkins et al., 2015*; *Tseng et al., 2015*; *Kizhatil and Bennett, 2004*; *Zhang and Bennett, 1998*; *He et al., 2014*; *He et al., 2012*; *Jenkins et al., 2013*). A neomycin resistance cassette, driven by the phosphoglycerate promoter (PGK-neo) and flanked by flippase recognition target (FRT) sites, was inserted between exon 42 and the 5' LoxP site. The linearized construct was introduced into 129X1/SvJ ES cells by electroporation, and G418-resistant clones were screened using polymerase chain reaction (*Figure 1C*). One of the primers was anchored outside of the 5' homology arm to allow the detection of the FLEx cassette within the correct chromosomal location. ES cells selected for the presence of the FLEx cassette using G418 were injected into C57BL/6NHsd blastocysts. A high percentage of chimeric animals were obtained and bred to C57BL/6NHsd mice to produce heterozygous animals. Mutant mice were backcrossed to C57BL6/J mice (Jackson Laboratory) and were compared to C57BL6/J mice as WT controls. These mice have been deposited at the Jackson Laboratory (B6;129S6-Ank3tm1Pmj/J, JAX strain # 038118).

### Genotyping

To confirm correct genotypes, ear punch biopsies were derived from adult breeding animals and tail cuts from all pups used for OTC preparation, using standard PCR procedures. Tissue samples were digested (25 mM NaOH, 0.2 disodium EDTA), centrifuged for 1 min at 14,000 rpm, and incubated at 95° C. After addition of 40 mM Tris-HCl and subsequent centrifugation at 14,000 rpm for 2 min, DNA

was amplified from gDNA using the forward primer JDH71 AG-GFP-F 5'CTACAACCAATGGGGA TCGTTAAC'3 and reverse primer JDH72 AG-GFP-R 5'TTAGGAAGGAGAAATGGGTGAGAG3'.

## Surgical procedures

### Viruses and Cre-driver lines

All Cre viruses, Cre driver lines, and controls used in this study are summarized in *Supplementary file 1B*.

### Stereotaxic injection

Virus injections were performed following the protocol outlined in *Rozov et al., 2020*. Mice received injections of 200–300 nl virus solution (*Supplementary file 1B*) into the ventral hippocampus [antero-posterior (AP), –2.4 mm; mediolateral (ML), ± 2.6 mm; dorsoventral (DV), –3.6–4.0 mm], 70–200 nl into the dorsal hippocampus [AP, –2 mm; ML, ± 2 mm; DV, –2.9–2.5 at a rate of 200 nl/min], 200 nl into prefrontal cortex [AP, 1.96 mm; ML, –0.35 mm; DV, –2.0 mm] or 300 nl into the primary motor cortex [AP, 0.3 mm; ML, –1.5 mm; DV, –0.7 mm]. After surgery, the animals were monitored during recovery and returned to their home cages. Virus expression and recovery were ensured for 3 wk, before perfusion or implantation of cranial windows.

### GRIN lens implantation

For deep brain imaging, 1 wk after virus injection, a gradient refractive index (GRIN) lens (diameter: 0.6 mm; length: 7.3 mm, 1050-004597, Inscopix) was implanted during a second surgery (systemic analgesia (30 min pre-surgery): buprenorphine (0.1 mg/kg); local analgesia: Lidocaine (10 mg/kg) and Ropivacaine (Naropin, 3 mg/kg); anesthesia: isofluorane (5% induction, 1.5-2% maintenance); subcutaneous post-operative analgesia: Metacam (5 mg/kg)). A 0.8 mm diameter craniotomy was drilled (Kyocera) above the basolateral amygdala (BLA) and a small track was cut with a 0.7 mm sterile needle (22G, Terumo). The GRIN lens was then slowly advanced into the brain [AP, –1.74; ML, –3.57; DV, –4.3 mm], fixed to the skull with light curable glue (Loctite 4305, Henkel), and the skull was sealed with Scotchbond (3 M), Vetbond (3 M) and dental acrylic (Paladur, Kulzer). A stainless-steel head bar (custom-made) was attached to fix the animal during the two-photon imaging sessions.

### Cranial window implantation

For the motor cortex imaging, 1 wk after the viral injection, a glass window was implanted above the left motor cortex (anesthesia and analgesia, see above). A craniotomy was drilled above the motor cortex using a dental drill (OmniDrill35, 503599, World Precision Instruments; burr drill bit: C1.104.002, Bösch Dental GmbH). Next, a circular coverslip was gently placed in the craniotomy and the edge of the coverslip was sealed with dental cement (C&B Super-Bond, Sun Medical). A stainless head bar was attached next to the glass window to fix the animal during the imaging sessions.

## Culture and slice preparation

### Preparation of dissociated hippocampal neurons

Cultures of dissociated mouse hippocampal primary neurons were prepared from postnatal P0-P1 ank-G-GFP mice of either gender and cultured on glass coverslips coated with 100 µg/mL poly-ornithine (Merck KGaA) and 1 µg /mL laminin (BD Biosciences). Cells were grown in the presence of 1-β-D-Arabinofuranosyl-cytosin (Merck KGaA) at 37°C and 5% $CO_2$. Cultures were transduced with lentiviruses at 4-6 d in vitro (DIV) and processed for immunostaining at DIV 19.

### Preparation of organotypic tissue cultures

Hippocampal organotypic tissue cultures (OTC) were prepared from ank-G-GFP animals of both genders according to previously published protocols [68]. Briefly, P4-5 old mice were decapitated and brains were explanted in preparation medium (94% Minimum Essential Medium (MEM, Thermo Scientific), 25% HEPES 1M buffer (Thermo Scientific), 10% GlutaMAX (Thermo Scientific), 10% Glucose (Sigma), 0.1 mg/ml Penicillin/Streptomycin (Sigma)), maintained at ~17° C. The cerebellum was removed and discarded to leave the neocortex, hippocampus, and connecting areas intact. Tissue blocks were mounted on a sliding vibratome (Leica) and cut horizontally at 300 µm. Slices

were transferred to a fresh petri dish containing preparation medium and hippocampi with a part of neocortex were dissected out from surrounding tissue. One brain yielded approximately six OTC, three of which were then placed onto Millicell membrane inserts (Millipore/Merck) in six-well tissue culture plates containing 1 ml pre-warmed culture medium (42% MEM, 25% Basal Medium Eagle; Thermo Scientific), 25% Normal Horse Serum, heat-inactivated (Thermo Scientific), 25 mM HEPES buffer, 2 mM GlutaMAX, 0.15% $NaHCO_3$, 0.65% Glucose, 0.1 mg/ml Penicillin/Streptomycin. OTC were maintained at 35° C in 5% $CO_2$. 500 µl culture Medium was exchanged every 2 to 3 d.

For AIS plasticity experiments, OTC were treated with either 6 mM KCl (Sigma) to achieve chronic stimulation, or 10 mM $MgSO_4$ (Sigma) to decrease spontaneous electrical activity.

## Preparation of ex vivo acute slices

After decapitation of the anesthetized animal, the brain was quickly removed and transferred to 4°C cold, carbogen buffered (95% $O_2$, 5% $CO_2$ at pH 7.4) artificial cerebrospinal fluid (ACSF) containing the following (in mM): 124 NaCl, 3 KCl, 1.8 $MgSO_4$, 1.6 $CaCl_2$, 10 glucose, 1.25 $NaH_2PO_4$, 26 $NaH_2CO_3$ with an osmolarity of 295 mOsm. The first third of the frontal brain and the cerebellum were removed. Horizontal brain slices were cut at 300 µm using a VT1000s or VT1200s Vibratome (Leica). The cutting solution contained (in mM): 140 potassium gluconate, 10 HEPES, 15 sodium gluconate, 0.2 EGTA, and 4 NaCl adjusted to pH 7.2 using KOH. Before recording, slices were incubated for 30 min in carbogen-buffered ACSF at 34°C for recovery and then kept at room temperature (~22°C) for at least 30 min before the start of experiments. For patch-clamp recordings, individual slices were moved into a submerged type recording chamber, restrained by a platinum-weighted harp, and constantly perfused with ACSF at room temperature (~22°C) at flow rates of ~3 ml/min.

## Staining procedures and histology
### Antibodies
All antibodies used in this study are summarized in *Supplementary file 1C*, including sources, specificity testing, and fixation protocols. Fixation and blocking reagents for all immunofluorescence experiments are summarized in *Supplementary file 1D*.

### Immunofluorescent staining
In this study, immunofluorescence was performed on (i) isolated hippocampal neurons, (ii) OTC, (iii) ex vivo acute brain slices, (iv) whole mount retina, and (v) cryosections from various brain regions. The basic staining protocol is the same for all tissue types, with a small adaptation of fixative, fixation times, and blocking buffer outlined in *Supplementary file 1D*.

After fixation, isolated hippocampal neurons were quenched for 5 min in PBS supplemented with 100 mM glycine and 100 mM ammonium chloride. Cells were permeabilized for 5 min in 0.1% Triton X-100, blocked with 1% BSA for 30 min, and incubated with primary antibodies diluted in PBS. PBS for 1-2 hr at RT or overnight at 4°C After washing in PBS, samples were stained with secondary antibodies and nanobodies 1-2 hr at room temperature or overnight at 4°C, washed and mounted in Mowiol supplemented with DABCO.

Acute slices, OTC, and tissues were fixed with 2 or 4% paraformaldehyde (PFA) at different times and with or without post-fixation (*Supplementary file 1D*). Cryosections were prepared from animals that were exsanguinated with 0.9% NaCl under deep anesthesia with Ketamine (120 mg/kg BW)/Xylazine (16 mg/kg BW) and perfusion-fixed with ice-cold PFA. Brains were then removed from the skull and were cryoprotected in 10% sucrose (overnight) followed by 30% sucrose (overnight) at 4°C. Tissue was trimmed to a block including the region of interest and embedded in Tissue Tek (Sakura Finetek).

After fixation, all samples were incubated in blocking solution (*Supplementary file 1D*) and incubated in primary antibodies over night at 4°C (exception: whole-mount retina, incubation in primary antibodies for 48 hr). After intensive washing (at least 3 × 10 min in PBS), samples were incubated in secondary antibodies for at least 2 hr in the dark. After 4 × 10 min washing in PBS, samples were mounted on glass coverslips using a water-based embedding medium with antifade properties (all immunostainings except for STED: Roti-Mount, Carl Roth, Karlsruhe, Germany; STED: Mowiol supplemented with DABCO, Merck, Darmstadt, Germany). All antibodies used in this study were previously verified and tested for specificity as outlined in *Supplementary file 1C*. In addition, negative controls were conducted for all stainings and consisted of omission of the primary antibodies; secondary

antibodies regardless of the species they were directed against did not produce any signals. All samples were stored at 4°C until imaging.

## Data acquisition, analysis, and statistics

### Laser scanning confocal microscopy

Confocal analysis was carried out either on a C2 Nikon confocal microscope (Nikon Instruments) equipped with a 60x objective oil immersion, 1.4 numerical aperture (NA), or a SP5 2MP (Leica Instruments) equipped with a 60x objective (Oil, 1.4 NA). Laser adjustments were optimized for each staining to achieve optimal signal-to-noise ratios. To increase the number of in-focus immunoreactive structures in some experiments, stacks of images were merged into a maximum intensity projection and saved as JPEG and TIF format. The thickness of single optical sections was 0.5 μm in stacks of at least 5-10 μm total depth and 0.25 μm in stacks of at least 2-5 μm, respectively. Representative images in all figures were optimized for contrast and brightness in Photoshop (Adobe).

### In vivo two-photon imaging

Somatosensory cortex: Implantation of cranial windows and two-photon imaging were performed following the protocol provided in *Knabbe et al., 2018*. Mice were single-housed after surgery and typically imaged starting 21 d after surgery. Two-photon imaging was performed with a TriMScope II microscope (LaVision BioTec GmbH) equipped with a pulsed Ti:Sapphire laser (Chameleon; Coherent). The ank-G-GFP signal was imaged at 920 nm wavelength using a 25x water immersion objective (Nikon, NA = 1.1). Anesthetized imaging sessions lasted no longer than 1 hr.

BLA and motor cortex: Two-photon images were acquired by combining two channels, green and red, either for a single plane image for BLA (250 frames averaged) or a z-stack along a 50 μm volume (1 μm step; 600 frames per plane, averaged online) for the motor cortex. using a custom-built two-photon microscope (Independent NeuroScience Services, UK). Mice were head-fixed with a custom-designed head bar system and placed on a running-wheel during imaging. The microscope was equipped with a resonant scanning system and a pulsed Ti:sapphire laser (Chameleon Vision S, Coherent). The microscope system was controlled by ScanImage software (Version 5.7, Vidrio Technologies). The ank-G-GFP and mCherry signals were imaged at 940 nm or 980 nm wavelength, and green and red fluorescent photons were collected through an objective lens (CFI75 LWD 16X W, 0.80 NA, Nikon). Photons were separated by a dichroic mirror (T565lpxr, long pass, Chroma) and barrier filters (green: ET510/80m, red: ET630/75m, Chroma), and measured by photomultiplier tubes (PMT2101, Thorlabs). The imaging frame was 512 × 512 pixels and images were acquired at 30 Hz. Due to the spatially non-uniform optical distortion that GRIN lens images produce (*Kitano et al., 1983*; *He et al., 2019*), it is challenging to produce correct scale bars for these images. Therefore, the scale bars indicated in *Figure 9E* are an estimate of the size of the image.

After acquisition, the images of each channel were merged and averaged to create dual-color images using ImageJ (Version: 2.0). 3D reconstructions of dual-color z-stacked images were created using the 3D Viewer plugin of ImageJ. Signal intensities in both green and red channels were adjusted in ImageJ for visualization purposes.

### STED microscopy and image analysis

Dissociated hippocampal cell cultures of ank-G-GFP pubs were prepared at P0 and infected at 4-5 DIV with lentiviral nGFP-Cre or nGFP-ΔCre vectors (expression through CMV promotor). Cells were imaged on an Abberior easy3D STED/RESOLFT QUAD scanning microscope (Abberior Instruments). STED lasers were 595 nm and 775 nm; excitation lasers were 485 nm, 561 nm, and 640 nm. Autocorrelation analysis was performed on 2 μm long regions and the profiles from single cells were averaged as described previously (*D'Este et al., 2017*). Autocorrelation amplitude analysis compares the difference between autocorrelation values at 190 nm and an average of values at 95 nm and 285 nm, respectively.

### 3D image reconstruction

Three-dimensional visualization of microglia, axon initial segments, and synaptopodin clusters were obtained by surface reconstruction of deconvoluted confocal images. All images were acquired under

close to ideal sampling density according to the Nyquist rate and further processed by blind iterative deconvolution based on the optical properties of the microscope and the sample using AutoQuant X3 (Media Cybernetics). To render x-y-z information of individual objects, surface structures were reconstructed using Imaris 9.8 (Bitplane, Zurich) according to previously published protocols (*Hanemaaijer, 2020*). Briefly, surface borders for each structure of interest are generated using a pixel-based image segmentation method that is based on a seed point detection algorithm. Cellular and subcellular structures can be selectively reconstructed based on their respective fluorescence signal, intensity, and diameter.

## AIS and noR length analysis

Based on confocal images, AIS and noR length analysis was performed using a self-written software tool (AISuite; *Roos and Engelhardt, 2020*; *Höfflin et al., 2017*; *Ernst, 2018*). Loading the original bioformat file into the software enables complete visualization of the three-dimensional expansion of the AIS and noR. Measurements were conducted by drawing a line through the AIS into the somatic domain as well as into the distal section of the axon and through the node, respectively. The ROI is then straightened along its vectors through affine transformations. The resulting transformation represents an AIS and noR, respectively, with surrounding tissue, straightened along its axis. A normalized intensity profile for the image channels is then calculated. For AIS, we defined the beginning and end as the first and respectively the last pixel value over a threshold of 40 % of the normalized fluorescence. The physical dimensions were assessed by converting the pixel data with the help of the meta data present in the bioformat file (here: 0.21 μm = 1 pixel): For AIS, the distance to soma was calculated as the distance from the start of the traced line to the start of the AIS, length as the extending line to the distal AIS. In total, 50 AIS per OTC per preparation (animal) and condition were quantified.

For noR, $GFP^+$ and $GFP^-$ noR were randomly chosen within 6 areas of white matter out of different samples, selected only based on ankyrin-G expression. A total of $n = 255$ nodes in nine images from three animals were quantified. Length measurements of nodes were conducted using the ankyrin-G (*Figure 3C*) or $Na_v1.6$ signal (*Figure 3—figure supplement 1B*).

## Electrophysiological recordings

In acute slices, CA1 pyramidal cells were visualized using an upright BX51 microscope (Olympus) with a 60x water-immersion objective (NA: 1.0). Recording electrodes were pulled using borosilicate glass on a Flaming/Brown P-97 Puller (Sutter Instruments) to yield a resistance of 3-6 MΩ. The electrode solution contained (in mM): 140 potassium gluconate, 3 KCl, 4 NaCl, 10 HEPES, 0.2 EGTA, 2 MgATP, and 0.1 $Na_2GTP$ (pH adjusted to 7.2, 288 mOsm). Ex vivo recordings were obtained in current-clamp mode with an ELC-03XS amplifier (NPI electronic). Signals were low-pass filtered at 3 kHz and digitized at 20 kHz using a Micros 1401MKII AC-converter (CED). Data were collected using the Signal 4.10 software (CED). Voltages were not corrected for the calculated liquid-junction potential of + 14.5 mV. Test pulses of –100 pA and 200 ms were applied regularly to control for changes in series resistance.

In isolated cultures, cells at DIV 16-20 were patch-clamped in whole-cell current-clamp configuration with a potassium gluconate-based internal solution solution containing (in mM): 140 K-Gluconate, 1 MgCl2, 10 HEPES-NaOH, 0.5 EGTA, 2 MgATP, 0.5 $Na_2GTP$, 10 Na-Phosphocreatin (pH adjusted to 7.4, 315 mOsm). The following extracellular solution was used (in mM): 125 NaCl, 2.5 KCl, 25 $NaHCO_3$, 0.4 Ascorbic acid, 3 Myo-inositol, 2 Sodium Pyruvate, 1.25 $NaH_2PO_4.2 H_2 0$, 2 $CaCl_2$, 1 MgClv, 25 D(+)-glucose (pH 7.4, 315 mOsm). In vitro recordings were performed using 3-4 MΩ electrodes pulled using borosilicate glass on a P100 Narishige Puller. Electrophysiological recordings were performed using a Multiclamp 700B amplifier (Molecular Devices) controlled with Clampex 10.4 software (signals were low-pass filtered at 2.5 kHz and digitized with 250 kHz). Data were analyzed offline using Clampfit 10.4 and custom-written macros in IgorPro (Wavemetric). APs were triggered by current injection through the recording pipette (500 ms, 25 pA steps from –200 to + 400 pA). Waveform analysis was performed at the first AP that fired 30 ms after the start of the current injection.

## Statistical analysis

Statistical analysis was performed using GraphPad Prism [8] (GraphPad Software, USA). STED data was first subjected to a Grubbs test (OriginPro [2020]) to identify outliers. Electrophysiology data were

analyzed in Signal [4.10] (CED, UK), MATLAB (The MathWorks, USA), or Igor (WaveMetrics Inc, OR USA) using custom routines.

Normally distributed data were tested by paired or unpaired t-tests (two groups) or ordinary one-way ANOVA followed by Tukey's multiple comparison test. Non-normally distributed data were analyzed using the nonparametric Mann-Whitney test (two groups) or Kruskal-Wallis one-way ANOVA followed by Dunn's multiple comparison test. Quantifications are given as median ± standard deviation. Box plots show median, 25%, and 75% percentiles (boxes). Significances are depicted in red font and as asterisks as follows: not significant (ns); $p < 0.05$ (*); $p < 0.01$ (**); $p < 0.001$ (***).

## Materials availability statement

Materials used in this study were obtained from commercial vendors, as detailed in **Supplementary file 1**, which includes full ordering information. The ank-G-GFP mouse line has already been distributed to multiple research laboratories. Researchers interested in accessing this mouse line are encouraged to contact the corresponding author for further details and availability. All images and electrophysiology files used in this study are openly available on Dryad.

## Acknowledgements

We are indebted to Katja Lankisch, Corinna Thielemann, Silke Vorwald, and Sabrina Innocenti for outstanding technical support in the conduction of these experiments. The authors acknowledge the Core Facility Life Cell Imaging Mannheim (LIMA) at the MCTN, Medical Faculty Mannheim, Heidelberg University, Germany, for providing the Imaris Workstation and SP5 confocal system (DFG-INST 91027/10–1 FUGG). This work was supported by the Deutsche Forschungsgemeinschaft, DFG (SFB 1134, TP03 and EN 1240/2–1 to ME, SFB1158-SO2 to CA), DFG Walter Benjamin Programme (PN 458054460 to CT), the European Research Council (Starting Grant 803870 to JG), Chica and Heinz Schaller Stiftung, 2019 NARSAD Young Investigator Grant, 2021 Fritz Thyssen Grant (to CA), the Swiss National Science Foundation (SNSF Professorship PP00P3_170672 to JG), and the Interdepartmental Training in Pharmacological Sciences training grant (T32GM140223) at the University of Michigan (to KMB). Jan Maximilian Janssen and Michael Bock are recipients of the MD Thesis Fellowship, Medical Faculty Mannheim, Heidelberg University.

## Additional information

### Funding

| Funder | Grant reference number | Author |
| --- | --- | --- |
| Deutsche Forschungsgemeinschaft | PN 458054460 | Christian Thome |
| Deutsche Forschungsgemeinschaft | EN 1240/2-1 | Maren Engelhardt |
| Deutsche Forschungsgemeinschaft | SFB 1134,TP03 | Maren Engelhardt |
| European Research Council | Starting Grant 803870 | Jan Gründemann |
| Chica and Heinz Schaller Stiftung | | Claudio Acuna |
| NARSAD Young Investigator Grant | | Claudio Acuna |
| Swiss National Science Foundation | PP00P3_170672 | Jan Gründemann |
| Interdepartmental Training in Pharmacological Sciences | T32GM140223 | Kalynn M Bird |

| Funder | Grant reference number | Author |
|---|---|---|
| National Institutes of Health | R01MH126960 | Paul M Jenkins |

The funders had no role in study design, data collection and interpretation, or the decision to submit the work for publication.

## Author contributions

Christian Thome, Conceptualization, Data curation, Formal analysis, Validation, Investigation, Visualization, Methodology, Writing – original draft; Jan Maximilian Janssen, Data curation, Formal analysis, Validation, Investigation, Visualization, Methodology, Writing - review and editing; Seda Karabulut, Nadja Lehmann, Masashi Hasegawa, Dan A Ganea, Chloé M Benoit, Formal analysis, Investigation, Methodology; Claudio Acuna, Formal analysis, Funding acquisition, Investigation, Methodology, Writing - review and editing; Elisa D'Este, Formal analysis, Investigation, Methodology, Writing - review and editing; Stella J Soyka, Formal analysis, Visualization; Konrad Baum, Michael Bock, Lia Y Min, Kalynn M Bird, Formal analysis; Johannes Roos, Software; Nikolas A Stevens, Formal analysis, Investigation; Jan Gründemann, Funding acquisition, Writing - review and editing; Christian Schultz, Resources; Vann Bennett, Conceptualization, Funding acquisition, Validation, Methodology; Paul M Jenkins, Conceptualization, Resources, Formal analysis, Supervision, Funding acquisition, Validation, Investigation, Methodology, Writing – original draft; Maren Engelhardt, Conceptualization, Resources, Data curation, Supervision, Funding acquisition, Validation, Investigation, Visualization, Methodology, Writing – original draft, Project administration

## Author ORCIDs

Christian Thome ⓘ https://orcid.org/0000-0002-3344-8683
Nadja Lehmann ⓘ https://orcid.org/0000-0003-4801-3057
Nikolas A Stevens ⓘ https://orcid.org/0000-0002-5592-3326
Lia Y Min ⓘ https://orcid.org/0000-0002-3977-1684
Paul M Jenkins ⓘ https://orcid.org/0000-0002-4207-5823
Maren Engelhardt ⓘ https://orcid.org/0000-0001-8020-6604

## Ethics

Ank-G-GFP mice were imported to Germany and a breeding colony maintained at the animal facility of the Medical Faculty Mannheim of Heidelberg University and subsequently shipped to and maintained by the Interfaculty Biomedical Research Facility in Heidelberg (IBF), Germany, and the University of Basel, Switzerland. All experiments were approved by the animal welfare officer of Heidelberg University and the cantonal veterinary office Basel-Stadt (3001 and 3058). Animal maintenance and experimental procedures were performed in strict accordance with German, Swiss, and US law, the guidelines of the European Community Council and the state government of Baden-Württemberg (Project T01/19, G-188/15, 35-9185/G-119/20, I-18/08, I-20/06). Ank-G-GFP mice were bred as a homozygote line and crossed to various Cre-driver lines as outlined below. Control animals were male C57BL/6N mice (Charles River Laboratories, Sulzfeld, Germany, Strain Code: 027) and mixed gender Thy1-GFP (Tg/Thy1-EGFP; MJrs/J provided by Jackson Laboratory Maine, USA). For experimentation, animals were 4-6-wk-old (20–25 g) and kept in groups of up to four individuals, with nesting material made of cellulose. Animals were maintained on a 12/12 hr light/dark cycle with controlled temperature and humidity, and access to food and water ad libitum. To minimize the stress of euthanasia, mice were sedated by exposure to 12-15% isoflurane until the animal fell unconscious.

Reviewer #1 (Public review): https://doi.org/10.7554/eLife.87078.3.sa1
Reviewer #2 (Public review): https://doi.org/10.7554/eLife.87078.3.sa2
Author response https://doi.org/10.7554/eLife.87078.3.sa3

---

# Additional files

## Supplementary files

Supplementary file 1. This Excel file contains detailed supplemental data supporting the findings

---

described in the manuscript. Sheet 1 A: Summary of statistics for passive and active properties (related to *Figure 7*). Sheet 1B: Summary of Cre viruses and Cre driver lines used in the study. Sheet 1 C: Specification of primary and secondary antibodies. Sheet 1D: Summary of fixation and blocking reagents used for all immunofluorescence experiments. Sheet References: List of references related to the materials provided in the supplementary file.

MDAR checklist

### Data availability

All data generated or analyzed for this study are included in the manuscript and supporting file. All images and electrophysiology files used in this study are openly available on Dryad.

The following dataset was generated:

| Author(s) | Year | Dataset title | Dataset URL | Database and Identifier |
|---|---|---|---|---|
| Engelhardt M | 2025 | Data from: Live imaging of excitable axonal microdomains in ankyrin-G-GFP mice | https://doi.org/10.5061/dryad.n02v6wx76 | Dryad Digital Repository, 10.5061/dryad.n02v6wx76 |

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
