## [Editor Report · eLife Assessment]

In this **valuable** paper, the authors created a reporter mouse line in which the Axon Initial Segment (AIS) is intrinsically labeled by an ankyrin-G-GFP fusion protein activated by Cre recombinase, tagging the native *Ank3* gene. Using confocal, superresolution, and two-photon microscopy as well as whole-cell patch-clamp recordings in vitro, ex vivo, and in vivo, the authors **convincingly** document that the subcellular scaffold of the AIS and electrophysiological parameters of labeled cells remain unchanged. They further uncover rapid AIS remodeling following increased network activity in this model system, as well as highly reproducible in vivo labeling of AIS over weeks.

---

## [Referee Report · Reviewer #1 (Public review)]

This paper introduces a new transgenic mouse line that allows the labelling of the AIS and nodes of Ranvier by tagging Ank-G with GFP in a Cre-dependent manner. The authors characterise the properties of the AIS and nodes of Ranvier when labelled with GFP to show that it has no adverse effects on the properties of the AIS and nodes of Ranvier, nor on most measures of intrinsic excitability in neurons. They also show that this mouse line can be used to follow AIS plasticity in vitro and to visualise the AIS of neurons in vivo. This is a very useful and timely tool that will make an important impact in the field.

---

## [Referee Report · Reviewer #2 (Public review)]

The axon initial segment (AIS) is the axonal domain where most neurons integrate inputs and generate action potentials. Though structural and electrophysiological studies have allowed to better understand the mechanisms of assembly and maintenance of this domain, as well as its functions, there is still a need for efficient tools to study its structural organization and plasticity in vivo.

In this article, the authors describe the generation of a knock-in mouse reporter line allowing the conditional expression of a GFP-tagged version of AnkyrinG (Ank-G), which is a major protein of the axon initial segment and the nodes of Ranvier in neurons. This reporter line can in particular be used to study axon initial segment assembly and plasticity, by combining it with mouse lines or viruses expressing the Cre recombinase under the control of a neuronal promoter. Furthermore, the design of the line should allow to preserve the expression of the main Ank-G isoforms observed in neurons and could thus allow to study Ank-G related mechanisms in various neuronal subcompartments.

Some mouse lines allowing the neuronal expression of AIS/node of Ranvier markers coupled to a fluorescent protein exist, however they correspond to transgenic lines leading to potential overexpression of the tagged protein. Depending on the promoter used, their expression can vary and be absent in some neuronal populations (in particular, the Thy-1 promoter can lead to variable expression depending on the transgene insertion locus). Furthermore, these lines do not allow conditional expression of the protein regarding neuronal subtypes nor controlled temporal expression. Finally, a thorough description of the in vivo expression of the tagged protein at the AIS, and its impact on the structural and electrophysiological properties of the AIS are missing for these lines.

The present reporter line is thus definitely of interest, as the authors convincingly show that it can be used in various contexts (from in vitro to in vivo). It could in particular be used to study the assembly and plasticity of the domains where Ank-G is expressed. The strength of this work is that it thoroughly characterizes the reporter line expression and shows that it does not alter the structural nor the electrophysiological properties of the labeled neurons. The additional data presented by the authors in the revised version adequately complete the previously shown data and address the questions raised by the reviewers.

---

## [Author Response]

The following is the authors’ response to the original reviews.

**Reviewer #1:**
R1-01 - Does ank-G-GFP label all isoforms (190, 270 and 480kDa) of ankG? From the images of the AIS and noR it appears that the large forms (270 and 480 kDa) are probably tagged with GFP. Did the authors check for puncta along dendrites and in dendritic spines, which are thought to be formed by the small (190 kDa) isoform? Perhaps a western blot to show that Ank-G-GFP labels all isoforms would be a useful addition to this study.

We believe that AnkG-GFP indeed labels the major Ank3 transcripts in the brain, including the 190, 270, and 480 kDa isoforms, based both on known mRNA exon usage and on Western blot analysis (data not shown). Thus, theoretically, this model would be useful for examining the localization of 190 kD ankyrin-G to dendritic spines. While we attempted to examine this in sections from tissue, it was difficult to separate punctate ankyrinG-GFP labeling from the background. However, these experiments were done in genetic crosses that would label most pyramidal neurons in a given area (i.e. CaMKIIa-Cre). Given the Cre-dependence of this model, future experiments could utilize sparse transduction with a Cre virus that also fills neurons with soluble fluorophores (i.e. mCherry or tdTomato) to mark isolated neurons and identify dendritic spines, as exemplified in Fig. 2D. This would allow examination of subcellular localization of ankyrin-G within single pyramidal cells before and after induction of synaptic plasticity.

R1-02 - In Figure 2, does all the native Ank-G get replaced by Ank-G-GFP? In Fig. 2E the GFP signal along the AIS of CamKII +ve neurons does not appear to be very homogeneous compared to the BIV-spectrin label. Have the authors carried out more experiments like those in 2F, using antibodies that label AnkG together with the GFP fluorescence of the labeled AnkG? It would also be informative to know if, as one might expect, the total levels of ankG-GFP correlate with the levels of ankG at the AIS.

We agree that this is an important point and conducted additional experiments to address your concerns. Of course, we cannot exclude that some unmodified ankyrin-G remains in the AIS or other structures. We expect the turnover of the protein to be rather slow, and native ankyrin-G likely remains to some degree. However, our quantification demonstrates that the ankyrin-G-GFP labeling is sufficiently homogeneous to accurately represent AIS size, indicating proportional levels of GFP to native ankyrin-G. Animals were crossed with a CaMKIIa-Cre driver line and ex vivo slices were imaged live and after immunolabeling. We found a strong correlation between live ankyrin-G-GFP (patch clamp chamber), postfix ankyrin-G-GFP, postfix ankyrin-G, and βIV-spectrin immunosignals of the same AIS. Furthermore, our measurements of AIS length using the intrinsic GFP signal in combination with ankyrin-G, or βIV-spectrin antibodies showed significant overlap (see R103). We now included these graphs as supplemental Fig. S2 in the manuscript (pp. 8-9, ll. 173-177).

R1-03 - Does the length and position of the AIS change when Ank-G is tagged with GFP? This seems like important information that is needed to make sure that there are no structural differences in AIS morphology when compared to native Ank-G.

This is a very important point. We used the βIV-spectrin signal to compare the length of AIS with and without GFP modification in acute slices after patch-clamp recordings (N = 3 animals, 27 GFP+ and 48 GFP- AIS). As secondary control, we plotted the measurements of 160 AIS from a Thy1-GFP mouse line (N = 3 animals, 160 AIS). We found no significant difference in the length and position of the βIV-spectrin signal between GFP positive and negative AIS (p=0.3364 unpaired t-test, p=0.6138 non-parametric Mann-Whitney test, respectively). We have now included this analysis as Supplemental Fig. S2A in the manuscript (pp. 8-9, ll. 173-177).

R1-04 - How was node length measured in Figure 3? Was this done using the endogenous ank-G signal? In this figure, it would be informative to also quantify the number of noRs with a Nav1.6 stain. Perhaps even check if there are correlations between Ank-G-GFP and Nav1.6 levels. In this figure, it appears that comparisons are carried out between Ank-G-GFP +ve and -ve neurons in the same cryosections, from Ank-G-GFP mice crossed with CamKIIa-Cre. I worry that this may not be comparing the same types of axons. What cells do the CamKIIa -ve axons belong to? Also, the labels on the bar graph are confusing - perhaps GFP+ve and GFP-ve would be clearer?

The reviewer raises an important point. We forgot to declare the signal which was used to measure node length in the manuscript. We have corrected this error and clearly state now in the Fig.3C legend that we used the ankyrin-G signal to quantify node length. Furthermore, using CaMKIIa-Cre mediated expression triggers ankyrin-G-GFP only in a genetically defined subset of neurons. Nodes that do not belong to this subgroup might very well have different node properties. Yet, we cannot assign potential differences in node length to the presence or absence of the GFP label, since we do not have an independent labeling technique for the very same subset of neurons. Since node lengths were similar and showed the same spread of lengths in our sample (Fig. 3C), we assume that the GFP length does probably not affect node length to a significant degree. We have now discussed this limitation in the result (p. 7, ll. 159-165) and method section (p. 30, ll. 644-645) and provide Supplementary Fig. S1 for more clarity. As suggested by the reviewer, we have measured mean fluorescence intensities between 91 GFP+ and 141 GFP- nodes using automated image processing in Imaris. The nodes were again defined by the ankyrin-G signal. We found no difference in length and ellipticity between the groups. We repeated this analysis and compared fluorescence intensities of Nav1.6 and ankyrin-G antibodies and again found no statistical differences between both groups. As suggested by the reviewer, we investigated whether ankyrin-G-GFP interferes with the fluorescence intensities of sodium channels (Nav1.6) and ankyrin-G in general. While the GFP signal showed a strong correlation with ankyrin-G, we found no interdependence with the Nav1.6 signal, indicating that the GFP label does not interfere with the general molecular composition of the nodes. We included these new analyses in Supplemental Fig. S1 (p. 7, ll. 159-165).

R1-05 - In Figure 4 it would also be important to show the distribution of AIS molecules along the AIS, compared to the GFP signal, to establish whether this spatial arrangement of AIS-specific molecules remains intact. For example, Nav1.6 has been described as a more distally-located channel. As the authors point out, the example in A appears to show precisely this feature, but there is no quantification. The same applies to Kv1.2. This would also allow the authors to provide some quantification across multiple AISs, rather than just example images.

We agree that quantifying and comparing AIS-associated proteins would be informative. We measured the intensity profiles of Nav1.6 and Kv2.1 in neighboring AIS and found no preferences for either end of the AIS, neither of GFP-positive nor GFP-negative AIS. We want to note that not all neurons exhibit a distal localization of Nav1.6 and hypothesize that our samples (neocortex layer II) also fall into this group. We included this new graph as Supplemental Fig. S2D and E in the manuscript (p. 9, ll. 180-184).

R1-08 - In Figure 4, did the +Cre condition result in all cells showing a GFP-labelled AIS? If not, were the autocorrelations for +Cre-treated neurons done specifically on cells that expressed AnkG-GFP?

We assume the reviewer refers to the autocorrelation in Figure 6. In this in vitro paradigm, we used virus-induced Cre expression which triggered ankyrin-G-GFP in almost all neurons. The orange boxplots describe the autocorrelation of all ankyrin-G, using a C-terminal antibody as in Fig.6C, but in neurons that also express ankyrin-G-GFP. The green samples use the GFP signal of ankyrin-GFP. We clarified this in the graph and legend of Fig. 6C (pages 14-15).

R1-09 - As mentioned above in Figure 3, the comparisons in Figure 5 (GFP +ve and -ve neurons) may not be comparing like-for-like neurons. I imagine that many of the CamKII+ve cells in the cortex and hippocampus will be GABAergic interneurons, whereas presumably all of the CamKII+ve neurons will be pyramidal cells. Have the authors made sure that they are comparing across the same cell types? The fact that the number of axo-axonic synapses is similar across the two populations (Fig. 5B) does suggest that similar neuron types (presumably pyramidal cells) were compared in the hippocampus, but some other way of making sure would be a nice addition.

We agree with the reviewer that the grey and green boxes are not sampled from the same subset of neurons, since only CaMKIIa-positive principal cells will express ankyrin-G-GFP. However, we are confident that the selected AIS belong to pyramidal neurons in both cases. Principal neurons can be well distinguished from interneurons not only by the size, shape, and position of their somas but also by the length and thickness of their AIS. We have performed previous studies on the AIS of interneurons using genetic GAD and parvalbumin markers. Thus, we are confident that the plots in 5A and 5B are sampled from pyramidal neurons, though certainly from genetically different subsets. We now highlight and discuss this limitation in the result section (p. 11, ll. 215-217) and modified the graph in Fig. 5A and 5B for clarity.

R1-10 - In Figure 6, what was the promoter for the DCre and Cre+ lentivirus? Was this also driven by CamKIIa? In culture it is not always easy to be sure of neuronal identity - did the authors try to bias their analysis to specific neuronal types?

Indeed, the nature of the promotor was not stated in the legend or method section, which we now corrected. We used lentiviral FUW-nGFP-Cre and FUW-nGFP-ΔCre constructs to trigger ankyrin-G-GFP expression. Both viruses use the CMV (Cytomegalovirus) promoter, which drives constitutively high levels of gene expression in a wide range of cell types, including neuronal cells. The majority of neurons in dissociated hippocampal cultures are excitatory, especially larger cells with larger AIS, which were preferably used in the analysis. Thus, we cannot claim that AIS nanostructure is intact in cultured interneurons, but this is also true for in vivo conditions in general. Since mice did not show any obvious behavioral phenotypes, we are positive that interneuron functionality is preserved. We also note that the parallel expression of nuclear GFP in the infected neurons was undesired, but did not impact STED imaging due to that technique’s high resolution.

R1-11 - The ability to visualize the plasticity of the AIS in real-time is an important advance in the field. The loss of proximal Ank-G-GFP signal upon local application of 15 mM KCl is particularly interesting. The fact that neighboring AISs are not affected is surprising - do the authors know how local their KCl application was? Also, although the neighboring AISs are a nice control, the one control lacking here is the local application of normal solution (preferably 15 mM NaCl to account for osmolarity changes) to make sure that this does not affect the properties of the AIS.

We used KCl puffs in previous, unrelated experiments where we observed that only cells directly in front of the pipette are visibly depolarized by an acute KCl puff (measured by patch-clamp). Due to technical limitations, patched and live imaged neurons were generally in the first 2-5 cell layers of the brain slice, which is well perfused by the constant flow of oxygenated ACSF. KCl is thus quickly diluted and carried away. We have visualized the concentration gradients via puff application by puffing the fluorescent marker fluorescein in the same recording condition. The cone of fluorescence was only visible in front of the pipette and vanished in less than a second post-pressure application. To verify that it is indeed KCl and not the mechanical stress that lead to the loss of proximal Ank-G-GFP, one would indeed need an ACSF puff control, which we did for other studies. However, this is not the point we wanted to make. Instead of studying live single-cell AIS plasticity, we want to demonstrate that such investigations are generally possible using the ankyrin-G-GFP line.

R1-12 - The ability to be able to image AISs in vivo is another important finding. Were the authors able to image noRs as well?

We believe that this is indeed the case. The panels in Figure 9C contain densely labeled puncta that also remain in position from week 1 to week 2. These are likely nodes of Ranvier, although we do not have the means to verify their presence at this time.

**Reviewer #2:**
R2-01 - Are there indeed different Ank-G-GFP isoforms expressed in this model and could they correspond to classical neuronal Ank-G isoforms?

This is an important issue that was also raised by reviewer #1. Please consult the respective section R1-01 above for our response.

R2-02 - What is the rationale of doing Ank-G co-labelling in the case of Ank-G-GFP expression, rather than Pan-Nav staining for example? The co-staining with Nav1.6 antibody, when present, is however convincing.

We used the co-labeling to emphasize that the ankyrin-G-GFP construct allows reliable investigation of the whole AIS. This is why we wanted to demonstrate that the ankyrin-G-GFP signal overlaps with other AIS markers, as well as all ankyrin-G in general (including potentially remaining native and unlabeled ankyrin-G). This was also a point raised by Reviewer 1, which is why we provided some additional graphs (see response R1-02). However, we agree that staining with another independent marker, such as Nav1.6 or βIVspectrin was necessary.

R2-03 - Figure 2D and F: what is the rationale for not using betaIV-Spectrin staining as in the other panels of this figure? Furthermore, could betaIV-Spectrin localization be affected by Ank-GGFP expression, as betaIV-Spectrin is known to depend on Ank-G for its AIS targeting? Are there any other AIS markers, which localization is known to be independent of Ank-G, that could have been used?

We have compiled this figure from a multitude of different experimental setups from different labs to showcase the reliability and robustness of the ankyrin-G-GFP label. This is why the type of staining is not consistent among panels. However, we provide some quantification on the possible impact of ankyrin-G-GFP expression on the βIV-spectrin signal and the composition of the AIS in general. The STED image verifies that the basic subcellular arrangement of the cytoskeleton, including βIV-spectrin, remains intact (Fig. 6). Most AIS markers are at least in some way dependent on ankyrin-G expression, but FGF14 and neurofascin may be the most independent candidates (Fig. 4).

R2-04 - Did the authors measure the mean AIS length and distance from cell soma in Ank-G-GFPexpressing neurons versus non-expressing ones (considering the same neuronal subtypes) to assess whether these were unaffected by Ank-G-GFP expression?

This is an important point that was also raised by Reviewer 1 (see also our comments to R1-03). We have included this analysis now in the manuscript as Supplemental Fig. S2A (pp. 8-9, ll. 173-177).

R2-05 - Figure 5C: the microglial staining and 3D reconstruction could have been clearer.

We have modified the image and 3D rendering to make Figure 5C clearer to the reader. We hope that our changes suffice.

R2-06 - Figure 8: do hippocampal neurons retain their electrophysiological properties after 20 DIV? It could strengthen this part of the work to have access to the electrophysiological data mentioned in the text.

This is an important issue. We did not perform any electrophysiological recordings in OTCs in the course of this study. Panel E uses acute hippocampal slices like in Fig. 7. We have performed patch-clamp experiments up to DIV 10 for an unrelated study (see graph for action potential firing, Author response image 2). There are not many studies performing electrophysiology in slice cultures due to the formation of a glial scar on top of the slices. However, multielectrode array (MEA) recordings demonstrated that hippocampal organotypic slice cultures remain viable and show electric activity past DIV 20 (though with decreased viability and activity). We kindly refer to the following publications on that matter:

**Author response image 2. sa3fig2:** Sample traces of action potentials triggered by current injections.

Gong W, Senčar J, Bakkum DJ, Jäckel D, Obien ME, Radivojevic M, Hierlemann AR. Multiple SingleUnit Long-Term Tracking on Organotypic Hippocampal Slices Using High-Density Microelectrode Arrays. Front Neurosci. 2016 Nov 22;10:537. doi: 10.3389/fnins.2016.00537. PMID: 27920665; PMCID: PMC5118563.

Mohajerani MH, Cherubini E. Spontaneous recurrent network activity in organotypic rat hippocampal slices. Eur J Neurosci. 2005 Jul;22(1):107-18. doi: 10.1111/j.1460-9568.2005.04198.x. PMID: 16029200.